# WRF-Simulated Low-Level Jets over Iowa: Characterization and Sensitivity Studies

Jeanie A. Aird[1], Rebecca J. Barthelmie[1], Tristan J. Shepherd[2], Sara C. Pryor[2]

[1]Sibley School of Mechanical and Aerospace Engineering, Cornell University, Ithaca, New York, USA
[2]Department of Earth and Atmospheric Sciences, Cornell University, Ithaca, New York, USA

*Correspondence to*: J. A. Aird (jaa377@cornell.edu)

**Abstract.** Output from six months of high-resolution simulations with the Weather Research and Forecasting (WRF) model are analyzed to characterize local low-level jets (LLJ) over Iowa for winter and spring in the contemporary climate. Low-level jets affect rotor plane aerodynamic loading, turbine structural loading, and turbine performance, and thus accurate characterization and identification is pertinent. Analyses using a detection algorithm wherein the wind speed above and below the jet maximum must be below 80% of the jet wind speed within a vertical window of approximately 20 m – 530 m a.g.l. indicate the presence of a LLJ in at least one of the 14700 4 km by 4 km grid cells over Iowa on 98% of nights. Nocturnal LLJs are most frequently associated with stable stratification and low turbulent kinetic energy (TKE) and hence are more frequent during the winter months. The spatiotemporal mean LLJ maximum (jet core) wind speed is 9.55 ms$^{-1}$ and the mean height is 182 m. Locations of high LLJ frequency and duration across the state are seasonally varying with a mean duration of 3.5 hours. Highest frequency occurs in the topographically complex northwest of the state in winter, and in the flatter northeast of the state in spring. Sensitivity of LLJ characteristics to the: i) LLJ definition and ii) vertical resolution at which the WRF output is sampled are examined. LLJ definitions commonly used in literature are considered in the first sensitivity analysis. These sensitivity analyses indicate that LLJ characteristics are highly variable with definition. Use of different definitions identifies both different frequencies of LLJs and different LLJ events. Further, when the model output is down-sampled to lower vertical resolution, the mean jet core wind speed height decrease, but spatial distributions of regions of high frequency and duration are conserved. Implementation of a polynomial interpolation to extrapolate down-sampled output to full-resolution results in reduced sensitivity of LLJ characteristics to down-sampling.

## 1    Introduction

The term low-level jet (LLJ) is applied to any lower-tropospheric (approximately 2 km or below) maximum of horizontal winds that exhibits confined vertical extent (Markowski and Richardson, 2011). LLJs are observed episodically in most regions of the world (Rife et al., 2010; Krishnamurthy et al., 2015). LLJ formation mechanisms and manifestations span a range of scales from synoptic (i.e. mid-latitude cyclones) down to meso- (i.e. weather fronts) and micro-scales (i.e. topographic complexity and day-night surface heating) (Blackadar, 1957; Chen and Kpaeyeh, 1993; Lackmann, 2002; Jiang et al., 2007; Tay, 2021). Mechanisms commonly invoked to describe the forcing mechanisms include diurnal (day-night) variations in baroclinicity over sloping terrain (referred to as the Holton mechanism, (Holton, 1967)) and diurnal variations in boundary layer friction (referred to as Blackadar mechanism (Blackadar, 1957)). Both mechanisms invoke decoupling of the planetary boundary layer from the surface. In the case of the Blackadar mechanism, this decoupling is due to changes in turbulent

mixing associated with day-night stability differences. These stability differences begin at sunset as the boundary layer rapidly stabilizes as the land surface cools, resulting in an inertial oscillation that is conducive to LLJ formation. For the Holton mechanism, the decoupling can be attributed to pressure gradients arising from day-night heating of sloping terrain. Thus, both mechanisms result in a wind speed maximum and indicate LLJs are most frequent under stable conditions and hence at nighttime (Holton, 1967), and in areas with topographic and/or land cover variability (Parish, 1982). LLJ characteristics, such as frequency, intensity and duration also vary by seasonal and inter-annual timescales (Weaver et al., 2009; Liang et al., 2015).

In the continental US, the Southern Great Plains (SGP) LLJ is a persistent and prominent warm-season climate feature manifest at the synoptic scale; it extends over multiple degrees of longitude (i.e. having a width of hundreds of kilometers) and is coherent over many degrees of latitude (i.e. the jet is oriented along a south-north axis parallel to the Rocky Mountains) (Weaver and Nigam, 2008; Rife et al., 2010). This jet is centered at heights below 850 hPa with a maximum (jet core, Figure 1) most commonly observed between 300-625 m height (Rife et al., 2010) and is associated with moisture flux and summertime precipitation (Higgins et al., 1997; Berg et al., 2015). Wind profiler observations at 250-m intervals from 500 m a.g.l. to 19 km from a network of 31 stations across the Great Plains suggested the mean LLJ height was approximately 1000 m and the mean duration was 2 to 4 hours (Mitchell et al., 1995).

LLJs are observed across a range of spatial and temporal scales and in both onshore and coastal environments. Observational data derived using minisodars and wind profilers deployed at the ABLE facility in Kansas in the US Southern Great Plains indicated the presence of southerly (72%) and northerly (28%) LLJs and the wind maxima typically occurred at 200-400 m a.g.l.. The southerly LLJs exhibited higher mean duration (~6.7 hours in the cold season and 6 hours in the warm season) than northerly jets (Song et al., 2005). As depicted in Figure 1, LLJs at, above and below these altitudes have the potential to impact the wind speed, turbulence, and shear across typical wind turbine rotor planes, and analyses of both observational data and WRF simulations indicate that LLJs frequently occur at heights that interact with the rotor plane (Gutierrez et al., 2014; Gutierrez et al., 2017; Nunalee and Basu, 2014; Wagner et al., 2019; Aird et al., 2020).

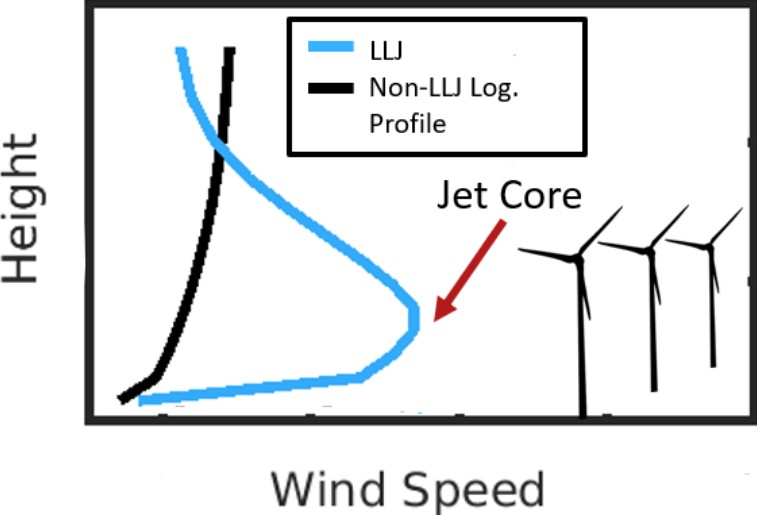

**Fig 1. Illustrative graphic comparing a low-level jet (blue) wind speed profile and a standard logarithmic, non-LLJ wind speed profile. The wind speed maximum (jet core) in this depiction is at a height below a typical rotor plane, which would result in negative shear across the rotor plane.**

Further, LLJs can increase wind farm performance through enhancing wake recovery (depending on atmospheric conditions and jet height), and may reduce wind turbine structural loading if the negative shear region of the jet interacts with the nacelle (Gadde and Stevens, 2021; Guttierez, 2017). If LLJ speed maxima occur at or near the

rotor plane, enhancements in turbulence and shear have implications for aerodynamic blade loading and longevity (Kelley et al., 2005). As wind turbine heights, rotor diameters, and capacities increase, it is likely that LLJs will interact more profoundly and frequently with the rotor plane, with increasing turbine dimensions resulting in more interaction with the jet core (Barthelmie et al., 2020).

Despite the pertinence of LLJ characterization to wind resources and wind turbine operating conditions, a

consistent and objective methodology for identifying and characterizing LLJ events is lacking. LLJ detection algorithms based on wind speed profiles employ:

1) Combined criteria based on both the absolute wind speed maximum and the difference in wind speed above and below the jet maxima (Bonner et al., 1968; Whiteman et al., 1997; Song et al., 2005).

2) A minimum absolute threshold for the difference in wind speeds above and below the profile maximum

(Andreas et al., 2000; Banta et al., 2002).

3) A minimum threshold for wind speeds above and below the jet maxima defined as a percentage of the wind speed maximum.

4) A combination of (2) and (3), requiring both, or one of the two, thresholds to be met (Lampert et al., 2015; Baas et al., 2009).

Use of subjective and varying thresholds render inter-comparison of the frequency and/or intensity of LLJs across studies difficult. Adding to this ambiguity, some studies entirely lack a quantitative LLJ definition.

Variations in the resolution of observational data or model output used to identify LLJs also contribute to ambiguity, inconsistencies in characterization, and/or a lack of generalizability (Kalverla et al., 2019; Whiteman et al., 1997; Bonner et al., 1968). For example, two analyses by Bonner et al. in 1968 and Whiteman et al. in 1997

of LLJs in the same region used similar criteria but differed in that the second study added a fourth LLJ criterion based on enhanced vertical resolution of rawinsonde data (Bonner et al. 1968; Whiteman et al. 1997). This led to detection of LLJs with stronger wind speeds and lower wind maxima than were found in the initial study. Thus, due to frequent variation of LLJ definitions, it is pertinent to examine the types of LLJs (characteristics) that each definition extracts and the agreement between definitions. As LLJs occur due to atmospheric forcing on multiple

scales (synoptic, meso, micro), it is possible that their wind speed profiles are a consequence of atmospheric conditions during the time of their generation, and jet profiles might be more likely to be extracted by certain definitions depending on atmospheric conditions or topography. A greater understanding of jets extracted through definitions used throughout literature can thus reduce uncertainty in future studies and inform choice of definition.

Research presented herein uses output from a simulation conducted using the Weather Research and Forecasting

(WRF) model to characterize LLJ occurrence and characteristics. The specific WRF configuration (e.g. selection of the planetary boundary layer (PBL) scheme) and horizontal and vertical resolution has a clear impact on simulated flow within the atmospheric boundary layer. In general, despite these sensitivities, WRF has been demonstrated to exhibit skill in simulating LLJ events and the near-surface wind climate, although WRF has been shown to underestimate the magnitude of the LLJ maxima (Storm et al., 2008; Schepanski et al., 2015;

Vanderwende et al., 2015; Squitieri et al., 2016; Smith et al., 2018; Gevorgyan, 2018; Pryor et al., 2020a). Here,

we do not further explore these dependencies but rather analyse WRF output to (i) develop a seasonal LLJ analysis for a warm and a cool season in the contemporary climate over a region within the US with high wind turbine densities and topographic variability, (ii) quantify the dependence of the LLJ characteristics (frequency, intensity, duration) and rotor plane conditions to the precise criteria used to identify LLJs and (iii) investigate the impact of vertical resolution on LLJ characteristics using full resolution and down-sampled WRF output.

## 2 Methodology

### 2.1 WRF simulations

The Weather Research and Forecasting Model (WRF) is a mesoscale numerical weather prediction model that is widely used in wind energy assessment and forecasting applications, such as predicting the impact of climate change on wind power generation and creating wind energy production estimates offshore and onshore (Pryor et al., 2020b; Salvação and Soares, 2018, Prósper et al., 2019). A high-resolution WRF (v3.8.1) simulation is conducted using a nested domain where the outer domain (D01) spans 150 by 150, $12 \times 12$ km grid cells and encompasses much of the US Midwest, while the inner domain (D02), centered over Iowa, comprises 246 by 204 $4 \times 4$ km grid cells (Pryor et al. 2020c) (Figure 2). This horizontal resolution has been found to be most optimal when simulating nocturnal LLJs when compared to higher (and lower) resolutions (Smith et al., 2018). A time step of 72 seconds is used for D01, while the time step in D02 is 24 seconds. 57 vertical sigma layers are employed and there are 25 levels below approximately 530 m a.g.l. Below 250 m a.g.l., the vertical spacing is approximately 15 m. Analyses presented here use model output sampled once hourly (at the top of the hour) for December 2007 to May 2008, and thus consider over 4300 profiles for each grid cell within a sub-domain (D03) comprising 147 by 100 grid cells that encompasses the state of Iowa (Figure 2). Iowa was selected as the focus for this work due to the high density of wind turbines (nearly 11GW of installed capacity) (American Wind Energy Association, 2019) and observational research that has indicated a high frequency of extreme positive wind shear, which may be associated with LLJs (Walton et al., 2014). Key physics settings in the simulation presented here parallel those used in a similar study of the Orinoco LLJ over South America (Jiménez-Sanchéz et al., 2019); i.e. the Mellor-Yamada-Nakanishi-Niino (MYNN) 2.5 (Nakanishi and Niino, 2006) PBL scheme is used, along with the MM5 surface layer scheme (Beljaars, 1995), and the Noah land surface model (Tewari et al., 2004). The MYNN scheme is selected as it has been validated previously for WRF simulations in the Great Plains and shown to adequately model the PBL height when compared to observations (Zhang et al., 2020). Further, studies of LLJs in the Great Plains indicate that nocturnal LLJ characteristics may be less sensitive to the scheme employed than vertical resolution; the MYNN scheme has been shown to have minimal mean absolute error when simulating key jet core conditions, particularly with fine vertical grid spacing and a high model top pressure level such as that utilized in this simulation (50 hPa) (Smith et al., 2018, Jahn and Gallus, 2018). Note that in all analyses presented herein only wind speeds within the lowest 530 m of the atmosphere are considered. This implicitly limits the detection of LLJs to levels below that height.

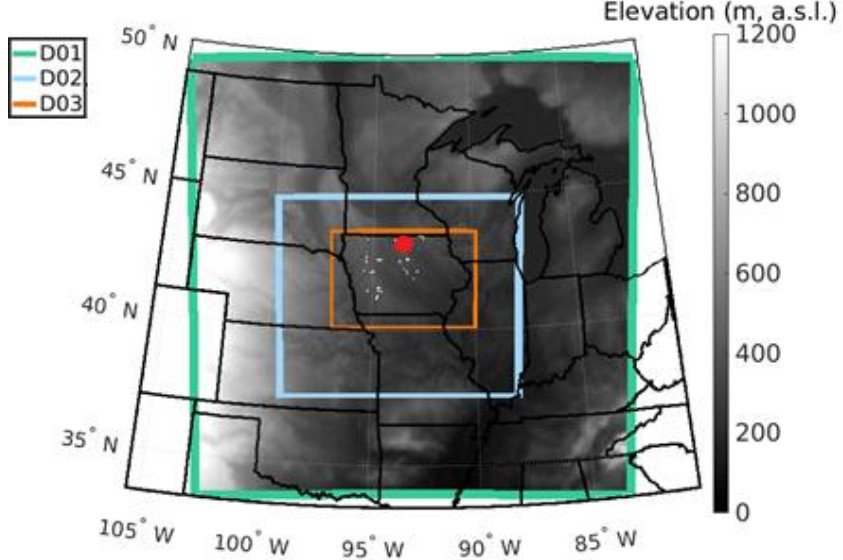

**Fig 2. Terrain elevation and domains used in the WRF simulation – D01, D02; and the region from which wind profiles are analyzed D03. White markers indicate wind turbine locations in 2014 ([https://eerscmap.usgs.gov/uswtdb/](https://eerscmap.usgs.gov/uswtdb/)). The red marker indicates the approximate location of the grid cell with highest LLJ frequency that is examined in Section 3.2.**

### 2.2    Seasonal Analysis: LLJ identification and meteorological conditions

The seasonal analysis of LLJ characteristics over Iowa is developed using a detection algorithm that employs a *variable* criterion of 20%, applied to WRF output for all grid cells. This detection algorithm means a LLJ is identified as present in a given profile if the wind speeds above and below the wind speed maximum have magnitudes that are at least 20% below the maximum (jet-core) wind speed. Thus, the threshold *varies* based on the maximum value in each wind speed profile. Cumulative density functions of atmospheric parameters

conditionally sampled based on the presence or absence of a LLJ are used to describe the conditions associated with LLJs. Parameters are considered in the vertical length of 50 to 150 m above ground level (a.g.l.), representing the rotor span of a typical wind turbine (not modelled here) with a rotor radius of 50 m and hub height of 100 m. The parameters considered are: (a) Mean turbulent kinetic energy (TKE) across the rotor plane derived by the PBL scheme. (b) Wind speed at a nominal hub-height of 100 m a.g.l. (c) The median Richardson number across

the nominal rotor plane ($Ri_{Rotor}$) specified as 50 – 150 m a.g.l (Eq. 1). (d) Mean shear ($\alpha$) across the nominal rotor plane (Eq. 2).

$$Ri_{Rotor} = \frac{2(Z_2-Z_1)g}{\theta_{Z_2}+\theta_{Z_1}} \left[ \frac{\theta_{Z_2}-\theta_{Z_1}}{(u_{Z_2}-u_{Z_1})^2+(v_{Z_2}-v_{Z_1})^2} \right] \quad (1)$$


$$\alpha = \left( \frac{U_{Z_2}-U_{Z_1}}{Z_2-Z_1} \right) \quad (2)$$

Where: U, u, v, and θ represent wind speed U, wind speed components u and v, and virtual potential temperature, respectively, at height Z a.g.l. $Ri_{Rotor} \sim 0$ is indicative of near-neutral stability, $Ri_{Rotor} > 0.25$ indicates stable conditions, and $Ri_{Rotor} < 0$ indicates unstable conditions (Grachev et al., 2013).


The $Ri_{Rotor}$ is similar to the Bulk Richardson number (Stull, 1988) but describes the dynamical stability across the wind turbine rotor (Nunalee and Basu, 2014). $Ri_{Rotor}$ and wind shear are calculated across each sigma layer in the nominal wind turbine rotor plane (six sigma layers fall approximately within this range). Thus, positive and negative shear due to LLJs are described at multiple heights within the rotor plane. TKE is also calculated at each of the six heights within the rotor plane. Mean TKE and shear and median $Ri_{Rotor}$ are then calculated from these points to approximate the central tendencies of rotor plane characteristics during non-LLJ and LLJ events. All variables except $Ri_{Rotor}$ are computed using output sampled at an hourly time step, while $Ri_{Rotor}$ is computed using variables output at three hourly intervals.

Probability distributions for LLJ characteristics, including duration and the jet core height, are also examined. If a LLJ occurs in a grid cell, the cell is flagged for each hour of occurrence. To calculate duration, these flags are counted for each consecutive LLJ occurrence, representing the length of time in which output from a given grid cell indicates the presence of a LLJ.

### 2.3    Sensitivity analyses

Following development of the seasonal analysis, two sensitivity analyses are performed (Table 1). The first sensitivity analysis (A) examines the impact of different detection algorithms on the resulting LLJ analysis. LLJs are detected and characterized using both; (i) *fixed* criteria i.e. a difference in wind speed above and below the wind speed maximum quantified in absolute terms (Andreas et al., 2000; Banta et al., 2002). (ii) *variable* criteria i.e. a difference in wind speeds above and below the wind speed maximum expressed as a percentage of the wind speed maximum. Often, these two types of criteria are used in conjunction, requiring a fixed *or* variable threshold or a fixed *and* variable threshold to be met (Baas et al., 2009; Lampert et al., 2016). This study examines both definitions separately to define the LLJs extracted under both types of thresholds. The criteria are grouped into five classes based on strictness and usage in literature, from the least strict (1 ms$^{-1}$ fixed, 10% variable) to the strictest (5 ms$^{-1}$ fixed, 50% variable) (Table 2). Threshold strictness increases across groups in increments of 1 ms$^{-1}$ for fixed and 10% for variable. Criteria group 2 features definitions most commonly used in tandem or uniquely in previous LLJ studies (2ms$^{-1}$ fixed, 20% variable).

Sensitivity to the LLJ definition employed is first demonstrated irrespective of domain-wide variations in topography using the WRF grid cell with the highest LLJ frequency according to the seasonal study developed initially (92.2784°W, 43.7467°N). Results are presented in terms of the mean LLJ profiles and the marginal probability of LLJs produced by each criterion. From this, a relative frequency of disagreement is calculated between the two LLJ definitions in each criteria group, indicating how often definitions (for each level of strictness) identify different LLJ events (i.e. how frequently variable criteria identify LLJs when fixed criteria do not, and the converse).

After the initial sensitivity is demonstrated, distributions of LLJ magnitude, duration, and jet core height are compared across the entire domain for each LLJ detection algorithm. The domain-wide temporal LLJ frequency is compared for thresholds in criteria group 2 (2 ms$^{-1}$ fixed, 20% variable) to examine definition sensitivity across varying terrain for each criteria type.

**Table 1. Summary of the LLJ Sensitivity Studies A & B.**

| Sensitivity study | Outline and Purpose | LLJ Identification Criteria | Output vertical sampling |
| --- | --- | --- | --- |

| A | Impact of different detection algorithms | 5 Variable and 5 Fixed thresholds (Table 2) | Full resolution |
|---|---|---|---|
| B | Vertical resolution of wind speed output down-sampled | 20% Reduction in wind speed above and below LLJ WS maximum | Full, half down-sample, quarter down-sample |

**Table 2. Criteria Groups for Sensitivity Study A and LLJ extraction algorithm thresholds.**

| Criteria Group | 1 | 2 | 3 | 4 | 5 |
|---|---|---|---|---|---|
| Fixed Criterion Threshold (ms$^{-1}$) | 1 | 2 | 3 | 4 | 5 |
| Variable Criterion Threshold (% of maximum LLJ wind speed) | 10 | 20 | 30 | 40 | 50 |

Sensitivity analysis B is conducted to examine whether, and by how much, LLJ characteristics change with the vertical resolution at which the WRF output is sampled. Wind speed output is down-sampled to a half and a quarter of the simulation resolution to investigate effects of wind speed profile data resolution when all other factors are unchanged. Results of this analysis are presented in terms of the spatiotemporal mean LLJ wind speed profiles, magnitude of the LLJs, duration, fraction of LLJs that impinge upon the rotor plane (defined as heights from 50-150 m a.g.l.) and the spatial patterns of LLJ frequency and duration.

## 3    Results

### 3.1    LLJ characterization using a variable threshold of 20%

A clear jet core is evident when comparing spatiotemporal mean LLJ and non-LLJ profiles normalized by each profile's respective wind speed maximum (Figure 3). The spatiotemporal mean LLJ core wind speed computed using wind speed values across each vertical layer for all hours from all grid cells is approximately 9.55 ms$^{-1}$ and is centered at about 183 m a.g.l. Approximately 96% of LLJs exhibit jet core wind speeds of 3-25 ms$^{-1}$ and are thus likely to be associated with normal wind turbine operation. Over the analysis period of six months there is evidence of a LLJ in one or more grid cells on nearly 98% of nights (between 8pm-6am local time) and nearly 65% of LLJs occur at night. Daytime LLJs are more frequent in the winter months (December - February). Approximately 40% of winter LLJs occur during daytime hours as compared to 30% during spring (March – May).

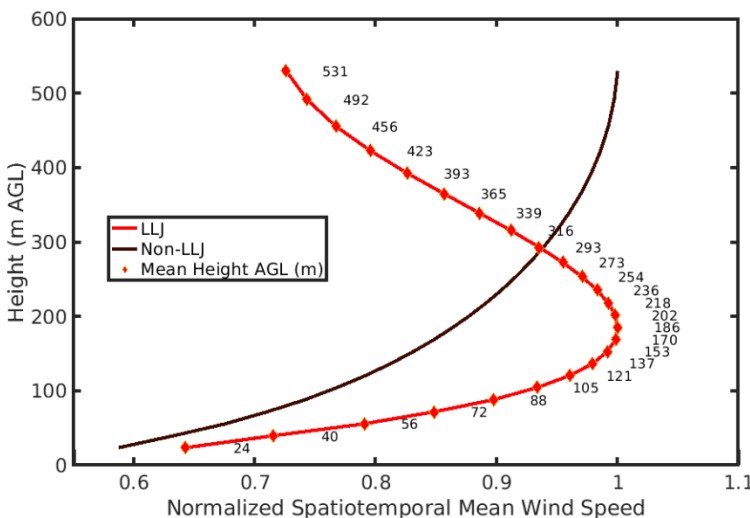

**Fig 3. – Mean wind speed profiles during all hours identified as exhibiting LLJs and those without (non-LLJ). These profiles are computed for all hourly profiles (in the entire time domain from December 2007 to May 2008) from all grid cells and each profile is normalized by the maximum wind speed after compositing. The LLJ detection algorithm uses a variable threshold of 20%. Both mean wind speed profiles are plotted against the temporally and spatially averaged mean height of each vertical level ( ◆ ).**

Thirty-percent of LLJs are evident only in individual hours, but 4% have a duration of > 10 hours (Figure 4(a)). The modal value of LLJ height in the vertical window considered is between 100-150 m a.g.l. (the upper extent of the nominal rotor plane), and approximately 39% of LLJs have a wind speed maximum within the nominal rotor plane of 50-150 m (Figure 4(b)).

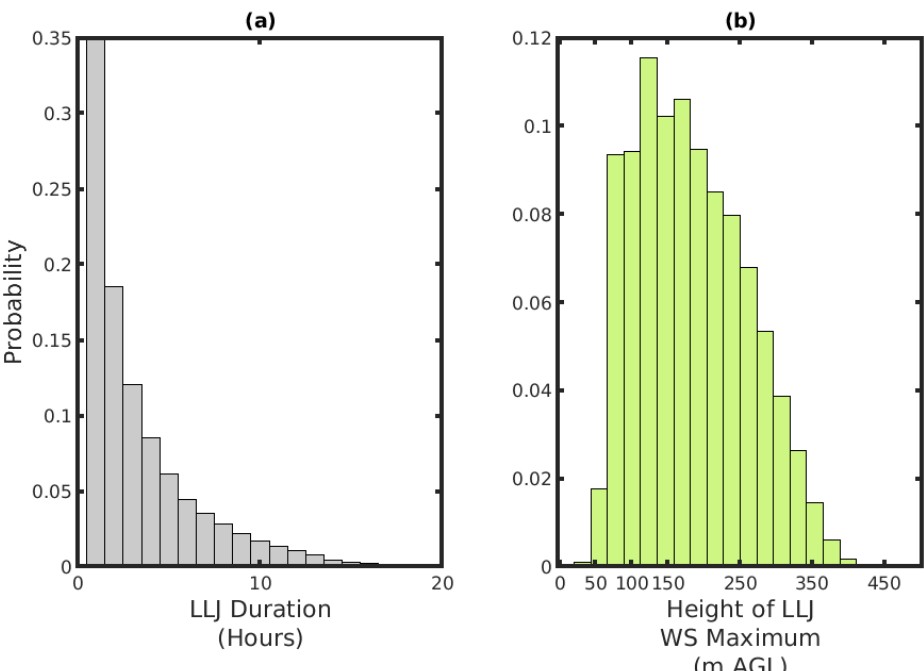

**Fig 4. – Probability distributions from a domain-wide sample of all hourly realizations (n=4392) of vertical LLJ wind speed (WS) profiles for: (a) LLJ duration; (b) Height of the jet core. Note that LLJ with durations of over 20 hours were identified, but the distribution is truncated at 20 hours for legibility.**

Consistent with expectations, LLJs are more prevalent during stable conditions as indicated by cumulative density functions of $Ri_{Rotor}$ , conditionally sampled by the presence or absence of a LLJ (Figure 5(a)). Approximately 15%

of LLJs occur during hours when $Ri_{Rotor} < 0.25$, but the spatio-temporal median $Ri_{Rotor}$ is 0.87 when the detection algorithm indicates the presence of a LLJ. Conversely, 60% of non-LLJ profiles occur with $Ri_{Rotor} < 0.25$, and the median non-LLJ $Ri_{Rotor}$ is 0.15. Also consistent with a priori expectations, LLJ events are associated with substantially lower TKE within the rotor plane. The median TKE within the rotor plane when LLJs are identified is 0.056 $m^2s^2$, while the non-LLJ median rotor plane TKE is 0.37 $m^2s^2$ (Figure 5(b)). Almost two-thirds (61%) of

LLJs exhibit wind speed maxima above the rotor plane. Thus, a greater diversity (i.e. wider distribution) of wind shear conditions occur during LLJs (Figure 5(d)), and there is evidence that very near-surface (i.e. low altitude) LLJs can induce negative shear across the nominal rotor plane (Gutierrez et al. 2017). Wind speeds at the nominal hub-height of 100 m a.g.l. are higher on average during non-LLJ conditions (Figure 5(c)), with a median of 9.24 $ms^{-1}$ when compared to the LLJ median of 8.02 $ms^{-1}$. This is likely due to a complex combination of the following

factors; (a) the LLJ selection criteria is more readily met at lower wind speeds (Section 3.2), (b) micro-scale to mesoscale features (i.e. locally forced LLJs) are less readily established under conditions with strong synoptic forcing that generates high geostrophic wind speeds (Mortarini et al., 2018) and (c) depending on the precise

height under consideration and the depth of the boundary layer, stable stratification may result in decreased vertical exchange of momentum (Barthelmie et al., 2013).

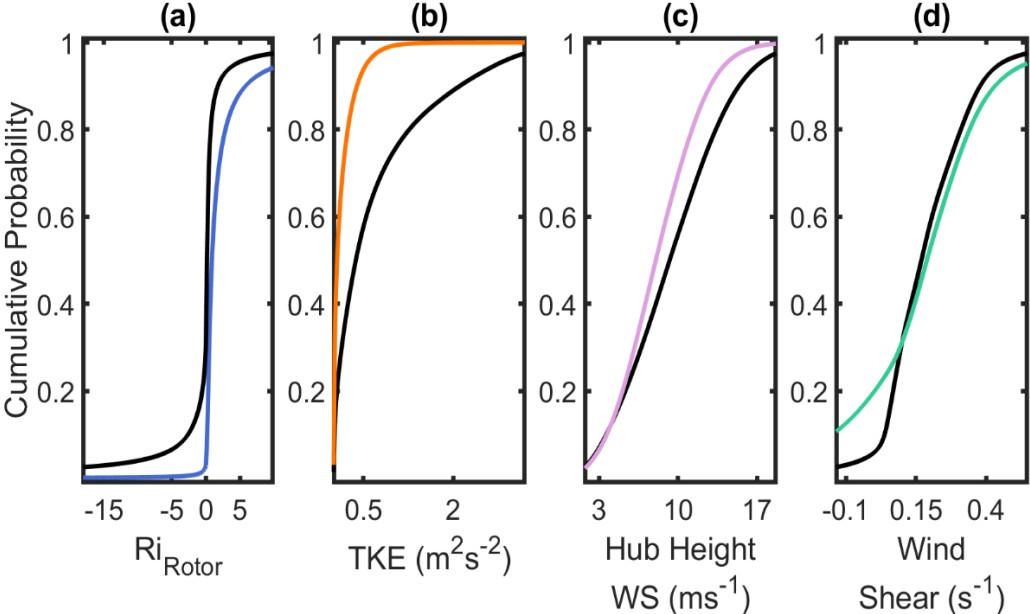


**Fig 5. – Domain-wide spatiotemporal cumulative density functions for conditions during hours with LLJ (colored) and without (non-LLJ) (black). Quantities (shear, TKE, and $Ri_{Rotor}$ ) are calculated at each of the six vertical layers within the nominal rotor plane (50 to 150 m a.g.l.) then averaged to obtain central tendencies (mean, median): (a) median $Ri_{Rotor}$; (b) mean TKE across the rotor plane; (c) hub height wind speed (wind speed at 100 m a.g.l.); (d) –mean wind**
**shear across the nominal rotor plane. For enhanced visibility, each subfigure is cropped at the 2.5th and 97.5th percentile values of non-LLJ parameters.**

The mean duration and frequency of LLJs exhibits a clear dependence on geographical location and season (Figure 6). On average, LLJs last slightly longer and occur more frequently in the winter months. The mean duration averaged over space and time is 3.6 hours in winter and 3.4 hours in spring. In spring, the northeast of Iowa
experiences the highest frequency of LLJs, with the detection algorithm using a 20% variable threshold detecting LLJs on up to 20% of hours. The mean LLJ duration in this season and region of Iowa approaches 4.5 hours. Conversely, the western part of the state is characterized by higher terrain elevation and larger terrain variability and exhibits a wintertime maximum of both LLJ duration and frequency (27% of hours) (Figure 6) consistent with formation of LLJs resulting from drainage-flow induced gravity waves (Prabha et al., 2011; Udina et al., 2013).

Mean wind vectors at a nominal wind turbine hub-height of ~ 100 m a.g.l. under LLJ and non-LLJ conditions suggest marked difference in both the mean wind direction in winter and spring and the mean wind directions (averaged in polar space) associated with LLJ and non-LLJ conditions (Figure 6(a) and (b)). The mean winter flow direction for both LLJs and non-LLJs exhibits a westerly component for all grid cells considered, while easterly flow components are more common during the spring months. Rotor plane wind directions during LLJ
events exhibit more spatial variability than during non-LLJ events. Springtime LLJs exhibit less spatial variability in wind direction than winter LLJs, coinciding with the increased frequency of winter LLJs compared to spring LLJs. Springtime LLJs are most frequently associated with northeasterly flow over the northeast of the state, while winter LLJs are most frequently associated with southwesterly flow in the northwest of the state. Analyses of the seasonality and spatial variability of mean LLJ wind directions indicate that, during winter over the western
portion of the state, LLJs are predominantly associated with southerly wind directions, while over eastern Iowa the LLJs are associated with more northerly flow (Figure 6(a)). Conversely, springtime LLJs over almost all of

the state are dominated by easterly wind directions and are generally of substantially shorter duration over the western half of Iowa (Figure 6).

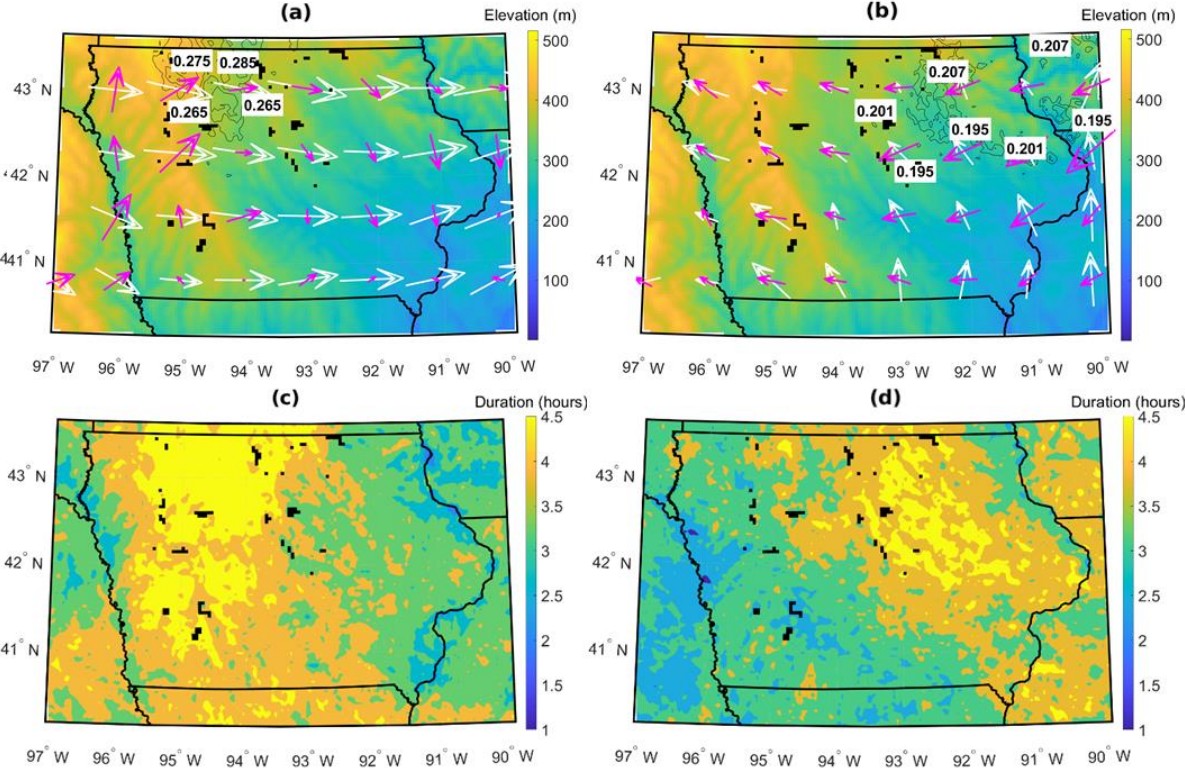

Fig 6. – (a) – Dec-Feb. Regional elevation (m) with contours of regions of highest 10% of LLJ frequency (>.26). Average LLJ (—) and non-LLJ (white) wind vectors at nominal turbine hub height of 100 m; (b) – Mar-May. Regional elevation (m) with contours (black, contour values given in white markers) of regions of highest 10% of LLJ frequency (>.19). Average LLJ and non-LLJ wind vectors at nominal turbine hub height of 100 m; (c) – Dec-Feb. Regional mean LLJ duration; (d) – Mar-May. Regional mean LLJ duration. Black markers indicate wind turbine locations.

This variation in LLJ intensity and duration by season and location may reflect differences in LLJ genesis mechanisms. The western portion of Iowa exhibits substantially more complex terrain and thus may be subject to stronger thermal (radiative) and dynamic forcing at the meso- and micro-scales. Consequently, this region may be subject to density-driven slope and valley winds that may induce LLJs via the Holton mechanism, particularly during winter (Holton, 1967). The increase of LLJ frequency in the northeast during the spring is also associated with an increase in LLJ speed when compared to LLJ wind speeds for the region in winter and may have a greater forcing contribution from the Blackadar mechanism (Blackadar, 1957).

### 3.2    Sensitivity analyses: LLJ detection algorithm

**i) Initial demonstration of sensitivity to LLJ definition**

Any LLJ analysis is naturally dependent on the detection algorithm applied. Thus, a sensitivity analysis is performed using differing LLJ detection thresholds (see Table 2). The impact of selecting different thresholds (five different fixed thresholds ranging from 1 to 5 ms$^{-1}$ in increments of 1 ms$^{-1}$ and five different variable thresholds ranging from 10 to 50% in increments of 10%) is illustrated in Figure 7 for the WRF grid cell that exhibited the highest LLJ frequency in the seasonal analysis (grid cell location indicated in Figure 2). Sensitivity is firstly demonstrated for a single grid cell to concisely prove sensitivity without confounding factors related to terrain elevation. Domain-wide frequencies are presented in Figure 9 for the most frequently used LLJ definitions

and indicate that there is terrain-related sensitivity to the LLJ criteria employed. Variable and fixed criteria in each group are studied separately to examine the type of LLJ extracted by each unique definition. In other words, in every case, *either* a fixed or variable criterion is applied; the criteria are not used in tandem throughout the study. As shown in Figure 7, the time-average mean wind speed profiles during hours identified as exhibiting LLJs using

these ten different selection criteria differ greatly. As the threshold used in the variable criterion increases, i.e. the difference between the LLJ core wind speed and the wind speeds above and below that level increases, the mean wind speed at the nominal wind turbine hub height and throughout the entire lowest 530 m of the model output decrease (Figure 7(a)). Conversely, as the fixed threshold for the difference in absolute wind speed of the jet core and above and below it increases from 1 to 5 ms$^{-1}$, wind speeds at the nominal wind turbine hub height and

throughout the entire lowest 530 m of the model output increase. These changes are non-linear and are most profound close to the mean height of the LLJ core (approx. 200 m a.g.l.). Alteration of the stringency of the threshold has a considerably more modest impact on the height at which the mean jet core is manifest (Figure 7). Application of increasingly stringent criteria (higher thresholds) causes the overall frequency of LLJs to decrease (Tables 3, 4). Interestingly, the absolute frequency of LLJs is approximately consistent for criteria groups across

the two methods (fixed and variable thresholds) (Tables 3, 4).

**Table 3. Marginal probabilities of LLJs when each of the fixed selection criteria are applied. Results are shown for hourly wind speed profiles from the single grid cell of highest LLJ frequency according to the seasonal study previously developed.**

| Criteria Group | 1 | 2 | 3 | 4 | 5 |
|---|---|---|---|---|---|
| Fixed Criterion Threshold (ms$^{-1}$) | 1 | 2 | 3 | 4 | 5 |
| LLJ frequency | 0.4110 | 0.2234 | 0.1116 | 0.0517 | 0.0198 |


**Table 4. Marginal probabilities of LLJs when each of the variable selection criteria are applied. Results are shown for hourly wind speed profiles from the single grid cell of highest LLJ frequency according to the seasonal study previously developed.**

| Criteria Group | 1 | 2 | 3 | 4 | 5 |
|---|---|---|---|---|---|
| Variable Criterion Threshold (% of maximum LLJ wind speed) | 10 | 20 | 30 | 40 | 50 |
| Variable: LLJ frequency | 0.4087 | 0.2336 | 0.0970 | 0.0326 | 0.0132 |

However, the mean wind speed profiles differ markedly. For criteria group 2, which features the fixed and variable criteria used (independently and in conjunction) throughout literature (20% variable/2 ms$^{-1}$ fixed), the temporal mean wind speed maximum for LLJ extracted with the variable criterion is approximately 4ms$^{-1}$ lower than that of the fixed (Hallgren et al., 2020; Andreas et al., 2000; Kalverla et al., 2019; Duarte et al., 2012).

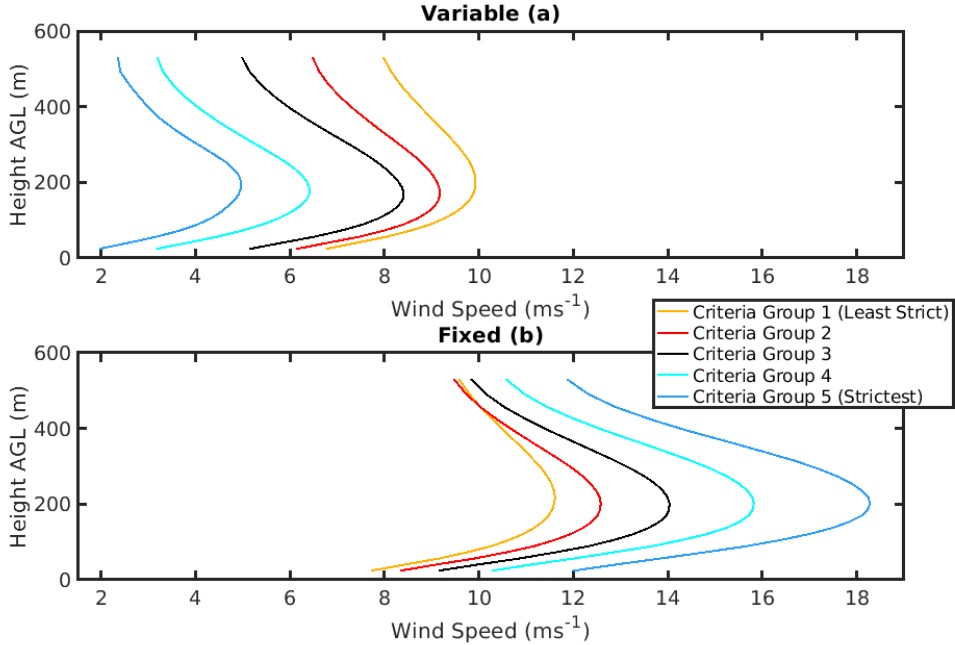

**Fig 7. – Temporal mean LLJ wind speed profiles extracted by each criterion (variable – (a) and fixed – (b)), colored by criteria group (criterion utilized for seasonal analysis is shown in red as part of Criteria Group 2). Temporal mean wind speed profiles per group are calculated from LLJ events as described in frequencies in Tables 3 and 4.**

Despite similarity in the frequency with which LLJs are detected as shown in Tables 3 and 4, the two criteria types (even in the least strict criteria group of 1 ms$^{-1}$ fixed, 10% variable) identify a substantial number of different, distinct LLJ events. For the least stringent criteria group (lowest thresholds), of the total number of times that a LLJ is identified between the two criteria (the intersection of identified LLJ), the criteria extract different LLJ events 20% of the time (i.e. a LLJ is identified by one type of criterion but not the other). Thus, the relative frequency of disagreement is 20%. This relative frequency of disagreement increases to nearly 40% for the second criteria group (2 ms$^{-1}$ fixed, 20% variable), in which the variable and fixed criteria identify different LLJ profiles 40% of the time (thus they identify the same hourly WS profiles as LLJs 60% of the time) (Figure 8). The frequency with which LLJs are identified by variable criteria but not by fixed, and vice versa, is relatively equal for the first three criteria groups. However, as threshold stringency increases (criteria groups 4 and 5), LLJs are more likely to be identified by fixed criteria than when the variable threshold is applied and the identified LLJ events become more dissimilar, with the two criteria identifying the same LLJ events only 10% of the time (Figure 8). These results indicate that the usage of varying LLJ definitions in literature (a fixed threshold only, or a fixed and variable threshold in tandem) potentially results in frequent identification of entirely different LLJ events. Results from this sensitivity study inform choice of criterion for the initial study; both criteria types are biased toward certain maximum LLJ speeds and choosing a criterion in the least strict group could result in LLJ wind speed profiles that are hardly differentiable from non-LLJ (as indicated by the lower shear displayed in jets extracted in criteria group 1). Further, criteria group 2 features definitions most relevant to previous studies, and the variable criterion chosen allows for analysis of LLJs that might have been previously undefined through usage of only a fixed criterion (as is common in previous literature).

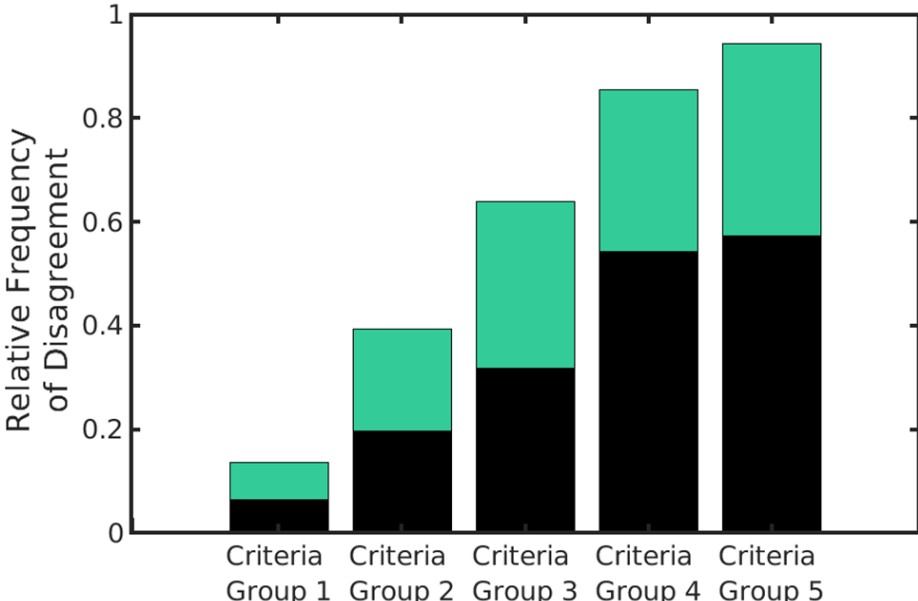

**Fig 8. – Relative frequency of disagreement of LLJ identification between analyses using a fixed threshold and a variable threshold. In each criteria group, the variable and fixed thresholds are applied separately to the same hourly wind speed profiles to generate frequencies of LLJ identification for each type of threshold. Bars represent the proportion of LLJ identifications in which one criterion identifies a LLJ while the other does not (the relative disagreement in LLJ identification between fixed and variable criteria). Bars are shaded by the proportion of disagreements in which: a LLJ is identified by fixed criteria but not variable (black), a LLJ is identified by variable criteria but not fixed (green). Calculated from hourly output from single grid cell with highest LLJ frequency as indicated by the seasonal analysis (see Figure 1 for location).**

### ii) Sensitivity of LLJ definition across entire domain (ensemble sensitivity)

Ensemble characteristics for LLJs extracted with each definition are analyzed to better understand LLJs extracted with each definition. Domain-wide LLJ frequencies are analyzed for the two most common definitions used in LLJ literature (criteria group 2) and indicate where, in a domain with complex terrain, each type of LLJ (as extracted by the definitions) is likeliest to be extracted. Results of the sensitivity analyses applied to all grid cells within D03 and all hours during the six-month period are consistent with those from the individual grid cell with highest LLJ frequency. Usage of a fixed threshold extracts LLJs with higher wind speed maxima overall; across all criteria groups, the ensemble median LLJ height is higher by approximately 20 m when fixed thresholds are applied (Figure 9(a)). Use of a higher variable threshold for LLJ detection (i.e. going from a deviation in wind speeds of 10% around the jet maximum to 50%) leads to a modest decline in the median height of the LLJ (Figure 9(a)) and a marked decline in LLJ duration from 6 hours to 2 hours (Figure 9(c)). Use of a stricter fixed threshold leads to an even smaller change in the median height of the LLJ maximum (Figure (9(b)). For all three properties, the LLJ cases become more self-similar (the dispersion of the distributions decreases) as increasingly selective criteria are applied (Figure 9). For all levels of strictness considered, variable criteria extract more cases that are identified as outliers (i.e. lie beyond 1.5 times the interquartile range from the 75[th] percentile) in terms of the LLJ duration than fixed criteria (Figure 9(c)). As in results for an individual grid cell shown in Figure 7, as the absolute threshold applied for LLJ detection increases, the LLJ maximum wind speed increases, whilst the converse is true for increasing the variable criteria threshold (Figure 9(b)).

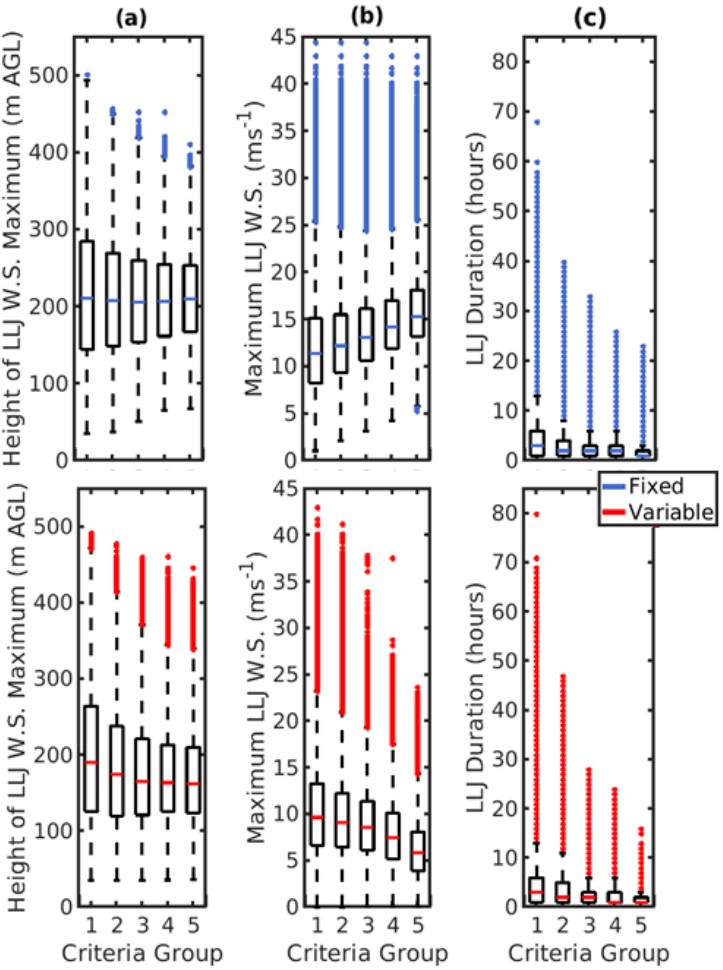

**Fig 9. – Box-whisker plots for definition-wise distributions of spatiotemporal LLJ characteristics (a) jet core height; (b) jet core speed; and (c) jet duration over the entire domain. Note: the whiskers on the boxplots extend from the 75th percentile to plus 1.5\* times the inter-quartile range, and from the 25th percentile to 1.5\* times the inter-quartile range. Points beyond those values are defined as outliers and plotted as individual points.**

For criteria group 2 featuring LLJ definitions commonly used in literature separately or in tandem, (2 ms$^{-1}$ fixed, 20% variable), the spatial distribution of LLJ frequency is sensitive to the threshold employed, particularly in regions of sloping and complex terrain (Figure 10). As illustrated by Figure 7 using output for a single grid cell, it is evident that algorithms using the two different criteria flag different periods as indicative of the presence of LLJs. The tendency for variable criteria to extract lower wind speed LLJs and for fixed criteria to extract higher speed LLJs is potentially evident in frequency differences between groups across varying terrain; for the area of high elevation in the west of the state, fixed criteria extract a higher frequency of LLJs than variable criteria on the western side of the terrain elevation. Conversely, on the eastern side, LLJs are extracted with higher frequency when a variable criterion is utilized. It is thus possible that variations in flow velocity over complex terrain contribute to the frequency differences in LLJs extracted by each criterion (Helbig et al., 2016). Areas with lower LLJ wind speed as defined in Figure 6 overlap with areas of higher LLJ frequency when a variable criterion is applied. The same is true for higher LLJ speeds when a fixed criterion is applied. The inference is that the two detection approaches, regardless of the precise thresholds applied, may exhibit differing ability to identify the presence of a LLJ depending on the causal mechanism, which has implications for regional LLJ studies in complex terrain.

Higher LLJ speeds in the surveyed region correspond to an atmosphere that is near-neutral and enhanced TKE (Aird et al., 2020). It is possible that a fixed criterion is more appropriate than a variable criterion to ensure that high speed LLJs are extracted reliably. Shorter duration, higher speed jets with enhanced TKE, such as those observed in higher frequency over complex terrain elevation, are less likely to be captured with the usage of a

410 variable criterion (Figure 10). In contrast, the variable criterion extracts a higher number of LLJ with low-magnitude wind speed maxima and higher duration. The decreased wind speeds of the LLJs captured under a variable criterion likely correspond to more stable conditions and decreased TKE. These characteristic differences further account for the higher frequency of LLJs extracted under a variable criterion in the region of the state with less complex and sloping terrain (Figure 10).

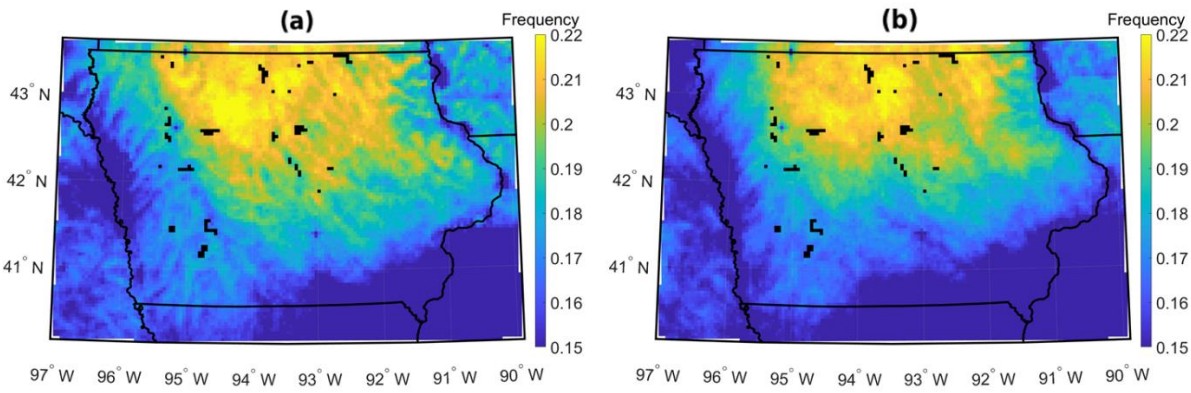

**Fig 10. – Spatial distributions of LLJ frequency computed using a detection algorithm with a (a) 20% variable threshold, (b) 2 ms$^{-1}$ fixed threshold.**

### 3.3    Sensitivity analyses: output resolution

In this analysis, a LLJ detection algorithm using a variable threshold of 20% is applied to output from the WRF

simulation using: the original vertical resolution, output sampled from every second level, and output sampled from every fourth vertical level (Table 3, Figure 11). The profiles are not linearly interpolated between vertical layers; the LLJs can only exhibit maxima at heights at the 25, 13, and 7 vertical layers considered (to parallel the extraction of LLJ profiles from observational data in which there are a number of fixed datapoints). The spatiotemporal mean LLJ core wind speed differs markedly according to the vertical resolution (Table 3). When

the model output is sampled at one-quarter of the simulation vertical resolution, the mean maximum (jet core) wind speed is 1 ms$^{-1}$ lower than when the LLJ detection algorithm is applied to output at the model resolution (i.e. all 25 levels below 531 m a.g.l.) (Figure 11, Table 3). Output down-sampled to one quarter resolution also exhibits a substantially lower mean LLJ core height (156.43 m) than when the analysis is applied to output at full resolution (182.64 m). This reduction in the height of the wind speed maxima results in a higher percentage of LLJ cores

falling within the nominal wind turbine rotor plane of 50 – 150 m a.g.l. The spatiotemporal mean duration and frequency of LLJs are also lower in the reduced resolution output (Table 3).

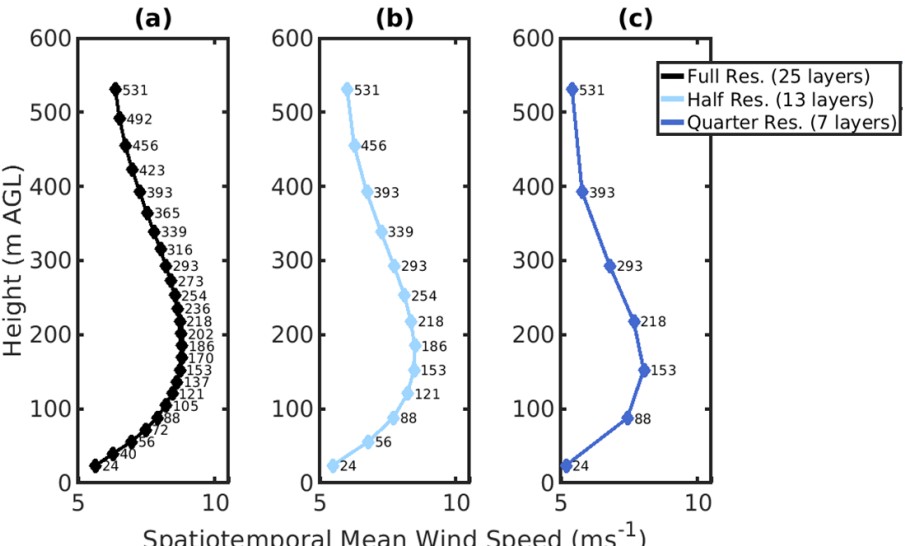

**Fig 11. – Mean wind speed profiles for output at; (a) – full resolution (25 layers, no down-sampling); (b) – half resolution (13 layers, output down-sampled to every other layer); (c) – quarter resolution (7 layers, output down-sampled to every fourth layer). Note: layers are connected linearly for figure visibility, but the LLJ wind speed maxima can only occur at the heights defined at the vertical layers (25, 13, and 7 heights respectively for each resolution).**

**Table 5. Spatially and temporally averaged LLJ properties as a function of model output vertical resolution.**

| | Mean Jet Core Wind Speed (ms⁻¹) | Mean Height of Jet Core (m a.g.l.) | Mean LLJ Duration (hours) | % LLJ with Jet Cores within the Rotor Plane | Spatiotemporal LLJ Frequency |
|---|---|---|---|---|---|
| Sensitivity Analysis B: Down-sampling of output | | | | | |
| Full Resolution: 25 Vertical Levels | 9.55 | 182.64 | 3.52 | 39.15 | 17.32% |
| 13 Vertical Levels (½ Resolution) | 9.18 | 172.89 | 3.35 | 41.83 | 15.12% |
| 7 Vertical Levels (¼ Resolution) | 8.53 | 156.43 | 2.98 | 46.95 | 10.75% |

The usage of a polynomial interpolation to account for lower output resolution when extracting LLJs is shown to reduce sensitivity in LLJ characteristics (Table 6). Winter wind speed output at full resolution is firstly analyzed for LLJs under the 20% variable criterion. From this, wind speed profiles corresponding with identified LLJs are sampled at quarter resolution (resulting in wind speed profiles comprised of 7 vertical layers). A sixth-degree polynomial is then fit to each of these wind speed profiles to extrapolate the non-linear LLJ shape between wind speed values at each layer. After creation of the polynomial, the quarter resolution height AGL for each profile is linearly interpolated to that of the full resolution output (25 layers). These linearly interpolated height values are then input into the polynomial function for each wind speed profile to extrapolate the quarter-resolution output into full-resolution output. These profiles (extrapolated to full resolution from quarter resolution) are then input into the LLJ detection algorithm (20% variable) and resulting ensemble characteristics are compared to LLJ characteristics from full resolution profiles and the original down-sampled quarter resolution profiles.

**Table 6. For winter months (Dec, Jan, Feb) - spatially and temporally averaged LLJ properties as a function of model output vertical resolution for full resolution and quarter resolution output, as well as quarter resolution output extrapolated to full resolution output through polynomial interpolation.**

|  | Mean Jet Core Wind Speed (ms$^{-1}$) | Mean Height of Jet Core (m a.g.l.) | % LLJ with Jet Cores within the Rotor Plane |
|---|---|---|---|
| Extrapolation to full-resolution output from quarter-resolution output | | | |
| Full Resolution: 25 Vertical Levels | 9.38 | 182.74 | 39.64 |
| Quarter Resolution Extrapolated to Full Resolution | 9.33 | 175.94 | 43.53 |
| 7 Vertical Levels (Quarter Resolution) | 8.34 | 158.15 | 50.73 |

LLJ characteristics (particularly jet core height) are sensitive to the model output resolution but spatial variability appears to be less sensitive. The temporal mean LLJ frequency and duration in each WRF grid cell, as extracted from quarter resolution and full resolution output, are normalized relative to their respective domain-wide maximum values (Figure 12). This process defines the domain-wide variations in LLJ frequency and duration for full resolution and quarter resolution output irrespective of the numerical values of each. The resulting normalized

LLJ frequency and durations for both resolutions allow for comparison of spatial variability. Most regions (irrespective of terrain elevation) display low sensitivity to reductions in resolution (Figure 12). Maximum positive and negative differences between normalized frequency and duration range from approximately -0.05 to 0.16, respectively. Regions of maximum spatial variability differences occur sporadically throughout the domain and do not correspond with terrain elevation. Regardless of these areas of high variability difference, the spatial

patterns of LLJ frequency and duration are comparatively insensitive to the down-sampling of vertical resolution for most of the domain. Further, regions identified as having the highest frequency and temporal mean duration (the highest 5% of each quantity) of LLJs are similar when the LLJ detection algorithm is applied to output at the original vertical resolution and one-quarter vertical resolution (Figure 12(a)). However, there is more divergence in spatial variation of LLJ duration than frequency when these contours are considered (Figure 12(b)). This

potentially indicates that inter-study comparisons of regions of high LLJ frequency (and less so duration) may be possible, even under reduced vertical resolution of observational data and/or model output.

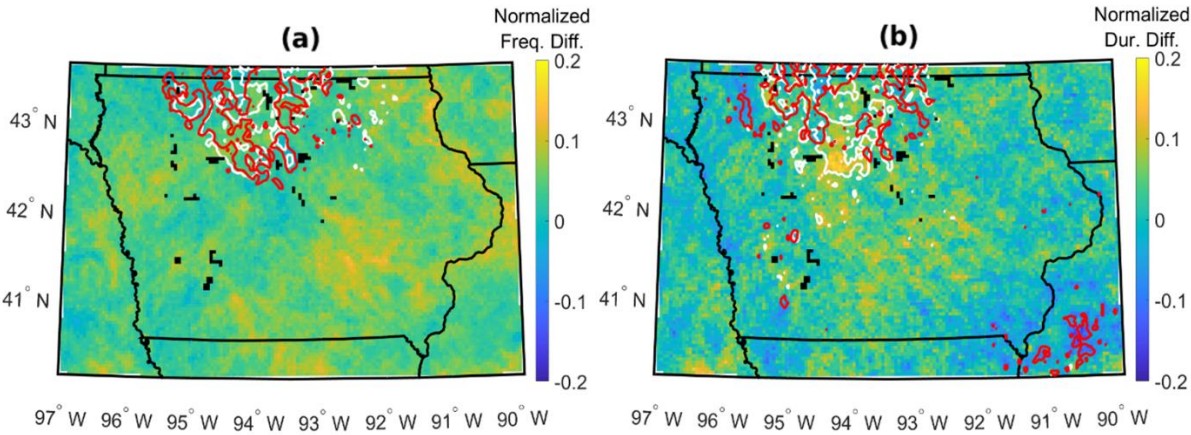

**Fig 12. –Mean spatial results for Dec. 2007 – May 2008, inclusive. Maps colored by difference in (a) normalized LLJ frequency and (b) normalized LLJ duration for output at full resolution and down-sampled to 7 layers. Contours**
**represent regions of highest 5% of (a) LLJ frequency and (b) LLJ duration for output at full resolution (white) and down-sampled to 7 layers (red).**

Ensemble LLJ characteristics display sensitivity to the resolution of wind speed profiles, but this can be mitigated through extrapolating the wind speed profile to higher resolution through a polynomial fit. This sensitivity appears to be consistent across the domain and irrespective of terrain complexity, as regions of highest LLJ frequency and

480 duration are preserved when LLJs are extracted from full resolution wind speed profiles and manually down-

sampled wind speed profiles. Though a 20% variable criterion is utilized for this sensitivity study, it is possible that usage of a different criterion might affect the results and increase the efficacy of the polynomial fit in resolving lower-resolution LLJ profiles. For example, for higher wind speed LLJs (wind speed maximum $> 17$ ms$^{-1}$) that are extracted by the fixed criterion, shear across the rotor plane remains relatively constant (Aird et al., 2020). In contrast, LLJs exhibiting lower wind speed maxima as are more commonly extracted by the variable criterion (wind speed maximum between 5 and 11 ms$^{-1}$) exhibit a nearly linear decrease in rotor plane shear with an increase in height A.G.L. These differences are attributed to lower jet core maximum heights for LLJs extracted with variable criteria (Figure 9). Thus, it is possible that extrapolating the LLJ profile from lower-resolution wind speed profiles as extracted from a fixed criterion would prove to be more effective due to more constant shear and higher wind speed maxima.

## 4    Conclusions

High resolution WRF simulations over the state of Iowa for December 2007-May 2008 are analyzed to generate a seasonal analysis of LLJs over the state and to assess the implications for wind energy resources and operating conditions. Properties considered are: maximum wind speed, height of the wind speed maximum, frequency, duration, and flow direction. Using a detection algorithm in which the wind speed above and below the LLJ must decrease by at least 20% of the jet core wind speed, approximately 95% of LLJs have wind speed maxima between 3 and 25 ms$^{-1}$ and the mean, modal and median heights of the LLJ core are approximately 183, 125, and 174 m, respectively. LLJs are found to be associated with low TKE across the rotor plane (50-150 m a.g.l.), to occur most frequently under stable conditions, and to cause comparatively high positive and occasionally negative wind shear across the rotor plane. LLJs are most common in the north of the state. Locations of highest regional LLJ frequency and duration are found to exhibit seasonal variability, likely due to changes in flow direction and the interaction between regional and locally forced flows.

Assessments of the sensitivity to the precise detection algorithm applied and output resolution are also performed. The first sensitivity analysis is conducted at full model output resolution and is designed to determine the sensitivity of LLJ characteristics to changes in LLJ definition. Two common types of criteria for LLJ definition are studied, labeled as *variable* and *fixed* criteria. Five criteria in each definition are considered (5 variable, 5 fixed) and are grouped by criteria strictness, ranging from 1 ms$^{-1}$ (fixed) to 10% (variable) for the least strict criteria group (criteria group 1), and 5 ms$^{-1}$ (fixed) to 50% (variable) for the strictest (criteria group 5). Sensitivity to LLJ definition is first illustrated for a single grid cell in the domain that exhibits the highest value of temporal LLJ frequency. Using different LLJ definitions is shown to identify not just different frequencies of LLJs but also different LLJ events. When considering all LLJs identified by the least strict criteria group, the definitions are shown to extract different LLJs for nearly 20% of the time. For the second criteria group that features LLJ definitions used in previous LLJ literature (2 ms$^{-1}$ fixed and 20% variable), the two definitions extract different LLJs (i.e. one definition flags a LLJ while the other does not) 40% of the time. This might suggest that combined criteria using a minimum fixed criterion of 2 or 2.5 ms$^{-1}$ combined with a 20 or 25% variable criteria will provide more robust results. Using output from all grid cells within the state of Iowa, it is shown that all LLJ characteristics are sensitive to changes in LLJ definition. LLJs extracted with each definition also likely differ in their causal mechanisms, as domain-wide sensitivities to the LLJ definition correspond to differences in terrain elevation and complexity. LLJs as extracted by fixed criteria are predominantly characterized by higher speeds and shorter

durations. LLJs extracted by a variable criterion exhibit a higher duration and lower wind speed maxima. In the context of previous work, lower LLJ wind speed maxima as extracted by variable criteria correspond to more stable conditions and decreased TKE, further explaining the increase in LLJ duration. The difference in LLJ types as extracted by each definition correspond to terrain complexity; in the region of the state with less complex and sloping terrain, a higher frequency of LLJs are extracted with the variable criterion. Previous literature implements

either a fixed criterion (most common) or a fixed and variable criterion in tandem. Thus, it is possible that for regions with less complex terrain, a variable criterion must be implemented to adequately capture all wind speed profiles with LLJ behavior. The converse is true for employing a fixed criterion: to adequately capture higher speed, shorter duration LLJs such as those that occur more frequently over complex and sloping terrain, it is pertinent to employ a fixed criterion. Thus, the usage of both a variable and fixed criterion to extract LLJs is

recommended. Future work to explore the impact of LLJ definitions in offshore conditions is warranted.

A second sensitivity study is conducted to determine the sensitivity of LLJ characteristics to changes in vertical resolution of the wind speed output. WRF output is down-sampled to one-half and one-quarter of the simulation resolution prior to application of the LLJ detection algorithm. All LLJ characteristics considered are found to be sensitive to reductions in wind speed profile vertical resolution but, as expected, characteristics calculated at ½

vertical resolution exhibit small percent differences from values at full vertical resolution when compared to those calculated at ¼ resolution, indicating that sensitivity to vertical resolution of wind speed data is non-linear. An implementation of a polynomial interpolation to extrapolate quarter-resolution output to full-resolution output is shown to reduce sensitivities of LLJ characteristics to the output resolution. While LLJ frequency and duration are sensitive numerically to output resolution, there is good agreement for the spatial variability of those

properties. These findings indicate that, while numerical values among LLJ studies may differ due to changes in wind speed profile vertical resolution, regions of high LLJ frequency may be correctly identified. Based on findings, employing a polynomial interpolation to enrich the number of datapoints in the wind speed profile may prove beneficial in resolving ensemble LLJ characteristics.

**Data availability.**

All of the hourly WRF output is available upon request from the authors via the DoE HPPS system.

**Author contributions.**

JAA, RJB and SCP jointly designed the analysis framework. JAA, RJB and SCP developed methods. JAA designed the sensitivity study analysis. JAA developed the figures, and drafted the initial paper with input from RJB and SCP. TJS performed the WRF simulations and SCP obtained the computing resources. SCP and RJB

also contributed to the writing of the final paper.

**Competing interests.**

The authors declare that they have no conflict of interest.

**Acknowledgements.**

The authors gratefully acknowledge the U.S. Department of Energy (DoE) (DE-SC0016438 and DE-SC0016605),
the National Science Foundation (NSF) Graduate Research Fellowship Program (DGE-1650441), and computing
resources from the NSF (ACI-1541215 and TG-ATM170024) and DoE (DE-AC02-05CH11231). The authors
also gratefully acknowledge the two reviewers for their insightful and helpful comments and suggestions.

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
