# Peer review of "WRF-Simulated Low-Level Jets over Iowa: Characterization and Sensitivity Studies"

_Wind Energy Science, 2020_

## Referee Comment (RC1) · Anonymous Referee #1 · 31 Dec 2020

Summary:

The manuscript use Weather Research and Forecasting (WRF) to predict the climatology of low-level jets (LLJ) in Iowa. Using Mellor-Yamada-Nakanishi-Niino 2.5 PBL scheme, MM5 surface layer scheme, and Noah land surface model, authors limited the study to winds in the lowest 530 m of the atmosphere. Specifically, they studied climatology parameters in the vertical space where the rotor of a hypothetical wind turbine (hub height=100 m, rotor radius=50 m) would spin.

In the second part, authors performed sensitivity analysis to assess how LLJ wind speed profile, duration and frequency are affected by the selection of different types of algorithmic LLJ definitions.

[Figure]

The paper presents a topic of high interest and potential practical use, especially to guide the location of future wind energy projects. This is particularly important for Iowa, a state that has been carrying an ambitious program of wind energy developments in recent years. However, the article in the present form still has some important issues to address. The use of some bulk parameters may introduce some uncertainties and merit at least a deeper explanation to demonstrate their merits. In some parts, the narrative is not clear and grammar could be improved. The article is in good track for publication at a later stage, but for now, I would recommend authors to address/respond the following comments:

Major comments:

1. Line 114 (equation 1), line 119 (equation 2), line 197 (figure 4): Are $z_2$ and $z_1$ equal to 150m and 50m (the maximum and minimum height of the turbine rotor) respectively? If that is the case, then the value of wind shear can be very coarse, especially if the core of the jet is right within that rotor interval. In that situation, a bulk wind shear would misleadingly suggest a smooth trend in wind speed from one point to the other, thus masking the existence of a strong positive wind shear in the lower portion plus a strong negative shear in the upper section. For the same reason, the Richardson number calculation wouldn't be too accurate to represent the transition in atmospheric stability (see comment 4).

2. Line 150 (table2): The grouping of both criteria types seems artificial. For example, it is clear that both the 5 m/s-criterion and the 50%-criterion leave out of the analysis many LLJs, but they do that at a different rate, with the 50%-criterion killing one third more LLJs (1-0.0132/0.0198). Both criteria have been grouped together despite having very different "strictness" power. Why not creating groups of criteria with similar "strictness" power (e.g., 5 m/s-criterion with 44%-criterion)?

3. Line 150 (table 2): Why table 2 gives marginal probabilities of LLJs in a single cell, instead of using the entire domain D3? It seems to me that finding marginal

probabilities in the entire domain studied would be more comprehensive in terms of taking into account different conditions of terrain, climatology, etc.

4. Lines 181-182: This 15% seems statistically significant and may indicate that the critical value of Ri_rotor (the transition between stable and unstable atmosphere) is different from 0.25 (but still in the positive). That may be another indication that Ri_rotor, being a bulk parameter, is not very precise as a measure of stability when calculated between a height below and a height above a LLJ peak. The function would be ill-conditioned due to the sensitivity of the shear term in the denominator. It would be interesting to know which fraction of the total jets detected are peaking within the rotor area.

5. Lines 190-191: LLJ's wind speed being lower than non-LLJ's wind speed is curious. Once the atmospheric layers are decoupled, the flow often accelerates to super-geostrophic wind, thus forming the LLJ. Hence, one would expect the wind speed at the heights of the jet core to be substantially greater than the wind speed at the same heights if there were no LLJ. One possible explanation is that, if the jet peak is happening outside (and above) of the turbine rotor (and probably the algorithm is detecting a fair amount of those cases), the rapid decay in wind speed downward due to stable stratification may lead to speed values inside the rotor area that are not so high. However, I am more inclined to accept the explanations provided later in the same paragraph. Explanations (b) and (c) are physically sound, but I am more inclined to think that the criteria used are missing some of the stronger jets (see comment 6). By the way, figure 6 (line 255) show mean wind speed that are much greater than 8.02 m/s when using fixed criteria. Weren't fixed criteria included to calculate the value shown here in line 191?

6. Line 255 (figure 6): The strictest fixed criteria (5 m/s) misses weaker LLJs because their peak wind speeds are not enough to have such speed decrease along the rotor radius (50m). Hence, fixed criterion's mean wind profile is biased upward because criteria discriminate against weaker LLJs. The strictest variable criteria (50%) misses

stronger LLJs because the wind speed decrease (e.g., 0.5x18=9 m/s) is too much to be observed within the limits of the rotor radius (50m). Hence, variable criterion's mean wind profile is biased downward because the criteria discriminates against stronger LLJs. The question is: if each criterion misses some LLJ incidents, why not use the least strict criteria (group 1) rather that group 2?

Minor comments:

1. Lines 73-74: To moderate expectations, it should be made clear that WRF historically has shown some shortcomings in modeling LLJs, with several studies showing WRF underestimating the maxima. The situation has improved in recent years, but LLJs have always been challenging to model with WRF.

2. Line 76: However, I would suggest to succinctly explain the merits that convinced you to use the specific schemes selected (schemes only mentioned in lines 94-96).

3. Line 88: "once" or "one"?

4. Lines 100-101: ". . .hub height. . . . . .nominal rotor plane. . .". If I understand correctly, there is no wind turbine modeled in the analysis. Presenting wind turbine's terms with no context may confound the reader as to where there is actually a wind turbine involved. I recommend to previously explain this. My personal suggestion would be something like: "Parameters are calculated in a vertical length (from 50 m to 150 m) where a hypothetical wind turbine (not modeled here) may spin, and hereafter we call that span the nominal rotor height, and the height 100 m, the hub height."

5. Lines 110-119: You may prefer to use a consistent style to enumerate a, b, c, d; either all of them in a single paragraph or each one in separated lines.

6. Lines 120-121: "All variables ... are computed at a disjunct hourly time step ... Ri_rotor is computed using output disjunct at three hourly intervals." Would you provide more details as to how and why time steps are "disjunct"?

7. Lines 130-133: "The five values used are 1:1:5 m/s. . . The five thresholds used

are 10:10:50%." One can infer that you mean "The five values from 1 m/s to 5 m/s in increments of 1" and "The five values from 10 % to 50 % in increments of 10" but the notation may be unclear to many. Is the notation supported by a standard?

8. Lines 168-171: May you rephrase Figure 2 caption? "...during hour identified as exhibiting LLJ..." seems to indicate that the red curve was obtained during a specific, single hour. However, the next sentence ("These profiles are computed for all hourly profiles from all grid cells...") points to something like an average profile using, not only several cells, but also from several hours. Moreover, I am curious as to how LLJs taking place in different grids and at different hours (and therefore potentially peaking at variable heights) were averaged into a unique profile. One can infer that you selected a specific hour in which calculations showed LLJ happening in several cells, then you combined the normalized profiles from those cells into an average profile (the heights of the LLJ's peaks should be very similar because they are happening in the same hour in not-so-distant locations), and finally did the same with the profiles in the grids with no LLJ happening to obtain the black curve. Is this interpretation correct?

9. Line 173: It is important to clarify that this modal value is obtained within the scope of this analysis (which only detected LLJs using wind speeds within the lowest 530 m of the atmosphere, as mentioned in section 2.1) and therefore cannot be interpreted as the modal value representing all LLJs in the region, which should also include LLJs peaking at higher altitudes. The modal value of all LLJs with core at any height would be probably higher.

10. Line 177: You may consider to spell out "WS" as "wind speed" if that is what it means. "WS" could also stands for "wind shear", for example.

11. Line 192: "see below". You need to be more specific as to where in the text you are directing the reader. Is it to section 3.2?

12. Lines 112-113: "The mean winter flow direction for both LLJ and non-LLJ is westerly," The arrows don't contrast much, but it seems that westerly flow direction is for

non-LLJs only, while LLJs exhibit much more spatial variability (Figure 5a).

13. Lines 113-114: "...while easterly flow is more common during the spring months." The arrows don't contrast much, but it seems that that easterly flow direction is for LLJs only, while non-LLJs seem to come mostly from south and southeast (Figure 5b).

14. Lines 212-220: Your cross-reference style is not consistent: Line 212: Figure 5(a) and (b). Line 218: Figure 5a. Line 220: Figure 5

15. Line 221 (figure 5): Would it be possible to use a more contrasting color for LLJ arrows in subfigures (a) and (b)?

16. Lines 222-224: If the color scale represents elevation and wind vectors are represented with arrows, then it is not clear which element in figures 5a and 5b is representing "contours of regions of highest 10% of LLJ frequency".

17. Lines 277-279: "The median LLJ height is higher by approximately 20 m when the fixed wind speed thresholds are applied than in use of any of the variable thresholds..." Revise sentence grammar.

18. Line 287: "...for applied for..." Check grammar.

Please also note the supplement to this comment:
https://wes.copernicus.org/preprints/wes-2020-113/wes-2020-113-RC1-supplement.pdf
* * *

---

## Referee Comment (RC2) · Anonymous Referee #2 · 20 Jan 2021

**1    General**

The paper presents an overview of LLJs in Iowa in winter and spring. The paper is well written, although a bit difficult to follow in some places. My main criticism is related to goal 1 and 3 (see line 76-80) of the paper:

**2    Major comments**

- Goal 1: to define a climatology one has to use at least a year of data and prefer-ably more (to capture all relevant mechanisms). The usual definition of a clima-

tological period is 30 years. Also in the context of wind energy the turbine life time is generally >20 years. It is quite likely that also summer time jets are quite abundant, if not more, than during winter and spring. For example in the cited paper of Baas et al. (2009), most LLJs were observed during summer. If you don't use a full year of data the paper is just a case study and in that case I don't think it contains enough novelty to publish the results.

- Goal 3: I agree with the paper that the detection could depend on resolution, but I was expecting to see a proposition of a method to help diagnosing the jet independent of resolution. At least something better than linear interpolation should be tested (see comment below).

**3 Minor comments**

- l7: I find it a bit confusing that the abbreviation LLJ is both used to indicate singular and plural. Maybe better to use LLJ for singular and LLJs for plural?

- l27: This is usually referred to as baroclincity, please add that term

- Table 2: It is not really clear to me whether these criteria are used seperately or together. If they are not used together, you should put them in seperate tables.

- l87: To represent a real climatological study one should at least cover all seasons.

- l231: This discussion would be much more interesting with some more physical interpretation. If you plot geostrophic wind speed and thermal wind speed in Fig. 5 it becomes clear if this mechanism plays a role here.

- l261: The explanation of this figure confusing and had to read this section several times to understand what was being plotted in Fig. 7. I am I correct that for group 2, approx. 60

- l294: "differs markedly" -> I can hardly distinguish any differences in Fig. 9. It would be more clear with a difference between the two plots, but also then I would probably not call it a marked difference. It seems it would be 1-2

- l328-330: This description is not very clear to me, maybe an equation would be better. So you normalize the wind speed profile by a maximum value in each grid cell and then calculate a frequency using the variable threshold and subtract those two frequencies? But then a difference of 0.1 is quite big, so it might be worth putting some more emphasis on that result in panel a?

- Fig. 10. This analysis requires the authors to use a simple polynomial fit or something similar to extrapolate the low-resolution case. Using a linear extrapolation in the points of the wind profile clearly does not reflect the non-linear behaviour of a LLJ profile.

- Conclusion: I was expecting to see some discussion on which method would be better or could be more suitable in certain conditions. The paper could benefit from a discussion section at the end of the results.

---

## Author Comment (AC1) · 4 Feb 2021

The authors are grateful for the reviewers' thoughtful and helpful comments and suggestions. We have responded to both reviews in the following pages and have edited the manuscript accordingly.

Review Responses – Review 1

Major Comments

1. Line 114 (equation 1), line 119 (equation 2), line 197 (figure 4): Are $z_2$ and $z_1$ equal to 150m and 50m (the maximum and minimum height of the turbine rotor) respectively? If that is the case, then the value of wind shear can be very coarse, especially if the core of the jet is right within that rotor interval. In that situation, a bulk wind shear would
misleadingly suggest a smooth trend in wind speed from one point to the other, thus masking the existence of a strong positive wind shear in the lower portion plus a strong negative shear in the upper section. For the same reason, the Richardson number calculation wouldn't be too accurate to represent the transition in atmospheric stability (see comment 4).

Response: $z_1$ and $z_2$ represent the change in sigma layer across the rotor plane; there are 6 sigma layers total that fall within the rotor plane (at approximately 50-150 m AGL). Each quantity (shear and RiRotor) is calculated across these layers, resulting in 5 values of shear and RiRotor across the rotor plane. TKE is output at every layer, and as such there are 6 values of TKE calculated across the rotor plane. The median RiRotor is then obtained from these sigma layer values in each LLJ and non-LLJ hourly wind speed profile; similarly, mean TKE and shear are calculated to represent the central tendencies of the rotor plane behavior for each variable. Thus, strong positive and negative shear across the rotor plane is accounted for in both the shear and RiRotor calculations. This has been clarified in the text in the following lines:

"The RiRotor is similar to the Bulk Richardson number (Stull, 1988) but describes the dynamical stability across the wind turbine rotor (Nunalee and Basu, 2014). RiRotor and wind shear are calculated across each sigma layer in the nominal wind turbine rotor plane (six sigma layers fall approximately within this range). Thus, positive and negative shear due to LLJs are described at multiple heights within the rotor plane. TKE is also calculated at each of the six heights within the rotor plane. Mean TKE and shear and median RiRotor are then calculated from these points to approximate the central tendencies of rotor plane characteristics during non-LLJ and LLJ events."

2. Line 150 (table2): The grouping of both criteria types seems artificial. For example, it is clear that both the 5 m/s-criterion and the 50%-criterion leave out of the analysis many LLJs, but they do that at a different rate, with the 50%-criterion killing one third more LLJs (1-0.0132/0.0198). Both criteria have been grouped together despite having very different "strictness" power. Why not creating groups of criteria with similar

"strictness" power (e.g., 5 m/s-criterion with 44%-criterion)?

Response: The criteria are grouped as such to draw parallels to criteria commonly used in LLJ literature. Although LLJ criteria differ by study, multiple studies have implemented those featured in the second group (2 ms-1 fixed and 20% variable). These criteria are often used in tandem or uniquely. Since these are the most frequent LLJ definitions (for onshore LLJs), they formed the basis of the criteria study and the other four groups were chosen in 1 ms-1 and 10% increments for continuity and to illuminate the differences in LLJs extracted by fixed and variable criteria.

Furthermore, a conclusion resulting from the grouping choice is also further clarified through the addition of the following:

"These results indicate that the usage of varying LLJ definitions in literature (a fixed threshold only, or a fixed and variable threshold in tandem) potentially results in frequent identification of entirely different LLJ events."

3. Line 150 (table 2): Why table 2 gives marginal probabilities of LLJs in a single cell, instead of using the entire domain D3? It seems to me that finding marginal probabilities in the entire domain studied would be more comprehensive in terms of taking into account different conditions of terrain, climatology, etc.

Response: The analysis begins with an initial demonstration of sensitivity to the LLJ criteria for a single grid cell; this was chosen to concisely prove sensitivity without confounding factors related to terrain elevation. Further, domain-wide frequencies are presented in Figure 9 for the most frequently used LLJ definitions; these indicate that there is terrain-related sensitivity to the LLJ criteria employed.

4. Lines 181-182: This 15% seems statistically significant and may indicate that the critical value of Ri_rotor (the transition between stable and unstable atmosphere) is different from 0.25 (but still in the positive). That may be another indication that Ri_rotor, being a bulk parameter, is not very precise as a measure of stability when calculated

between a height below and a height above a LLJ peak. The function would be ill-conditioned due to the sensitivity of the shear term in the denominator. It would be interesting to know which fraction of the total jets detected are peaking within the rotor area.

Response: An analysis of the LLJs occurring during hours when RiRotor <0.25 could be carried out in future work, although it is possible that this 15% occurs with higher LLJ speeds (as shown in our previous LLJ work in Iowa published through TORQUE 2020). Interestingly, in the vertical window considered, a sizeable proportion of LLJs (approximately 39%) peak within the rotor area (we show this later in Table 5). RiRotor and shear calculations are calculated across each vertical sigma layer in the rotor plane to increase precision (as discussed in response to #1).

5. Lines 190-191: LLJ's wind speed being lower than non-LLJ's wind speed is curious. Once the atmospheric layers are decoupled, the flow often accelerates to super-geostrophic wind, thus forming the LLJ. Hence, one would expect the wind speed at the heights of the jet core to be substantially greater than the wind speed at the same heights if there were no LLJ. One possible explanation is that, if the jet peak is happening outside (and above) of the turbine rotor (and probably the algorithm is detecting a fair amount of those cases), the rapid decay in wind speed downward due to stable stratification may lead to speed values inside the rotor area that are not so high. However, I am more inclined to accept the explanations provided later in the same paragraph. Explanations (b) and (c) are physically sound, but I am more inclined to think that the criteria used are missing some of the stronger jets (see comment 6). By the way, figure 6 (line 255) show mean wind speed that are much greater than 8.02 m/s when using fixed criteria. Weren't fixed criteria included to calculate the value shown here in line 191?

Response: We agree with this input – a lower wind speed at nominal hub height during LLJ conditions as opposed to non-LLJ conditions is likely due to the approximate 39% of LLJs that peak in the rotor plane and the definition employed, as well as explanations

in (b) and (c). Had the definition employed been fixed rather than variable, it is likely that the LLJ wind speed at the nominal turbine hub height would be biased toward stronger LLJs (as compared to the variable criterion, which is biased toward weaker LLJs). Fixed criteria were not included to calculate the value shown in line 191 – the criterion for the seasonal analysis is 20% variable. This is discussed further in the following paragraph (after Fig 3). 8.02 m/s is the median spatiotemporal LLJ wind speed maximum value, while the mean is 9.55 m/s. Profiles in Figure 6 display the mean wind speeds and are also for a single grid cell.

6. Line 255 (figure 6): The strictest fixed criteria (5 m/s) misses weaker LLJs because their peak wind speeds are not enough to have such speed decrease along the rotor radius (50m). Hence, fixed criterion's mean wind profile is biased upward because criteria discriminate against weaker LLJs. The strictest variable criteria (50%) misses stronger LLJs because the wind speed decrease (e.g., 0.5x18=9 m/s) is too much to be observed within the limits of the rotor radius (50m). Hence, variable criterion's mean wind profile is biased downward because the criteria discriminates against stronger LLJs. The question is: if each criterion misses some LLJ incidents, why not use the least strict criteria (group 1) rather that group 2?

Response: The algorithm searches the vertical window considered (up to 530 m AGL) for a decrease in speed sufficient to meet both types of criteria (so the algorithm is not just searching the rotor plane). We agree with this analysis – fixed and variable criteria are biased toward stronger and weaker LLJs, respectively. Each criterion misses some LLJ incidents, although this is reduced for the first criteria group. It is difficult to quantify which definition to use (although this may be explored in further studies) since both criteria types are biased toward certain maximum LLJ speeds and choosing a criterion in the least strict group could result in LLJ wind speed profiles that are hardly differentiable from non-LLJ. Further, criteria group 2 features definitions most relevant to previous studies, and the variable criterion chosen allows for analysis of LLJs that might have been previously undefined through usage of a fixed criterion. We have tried

to emphasize this important point by repeating in the Conclusions:

"Using different LLJ definitions is shown to identify not just different frequencies of LLJs but also different LLJ events. When considering all LLJs identified by the least strict criteria group, the definitions are shown to extract different LLJs for nearly 20% of the time. For the second criteria group that features LLJ definitions used in previous LLJ literature (2 ms-1 fixed and 20% variable), the two definitions extract different LLJs (i.e. one definition flags a LLJ while the other does not) 40% of the time."

And we have added also the following to the abstract:

"Use of different LLJ definitions identifies both different frequencies of LLJs and different LLJ events."

Minor Comments

1. Lines 73-74: To moderate expectations, it should be made clear that WRF historically has shown some shortcomings in modeling LLJs, with several studies showing WRF underestimating the maxima. The situation has improved in recent years, but LLJs have always been challenging to model with WRF.

Response: This has now been acknowledged in the introduction, and in context with the previous sentence reads:

"The specific WRF configuration (e.g. selection of the planetary boundary layer (PBL) scheme) and horizontal and vertical resolution has a clear impact on simulated flow within the atmospheric boundary layer. In general, despite these sensitivities, WRF has been demonstrated to exhibit skill in simulating LLJ events and the near-surface wind climate, although WRF has been shown to underestimate the magnitude of the LLJ maxima (Storm et al., 2008; Schepanski et al., 2015; Vanderwende et al., 2015; Squitieri et al., 2016; Smith et al., 2018; Gevorgyan, 2018; Pryor et al., 2020a)."

2. Line 76: However, I would suggest to succinctly explain the merits that convinced you to use the specific schemes selected (schemes only mentioned in lines 94-96).

Response: A sentence has been added to clarify usage of the PBL scheme, which has been shown to affect LLJ characteristics:

"The MYNN scheme is selected as it has been evaluated previously for simulations of the LLJ in the Great Plains, which indicate that LLJ characteristics may be less sensitive to the scheme employed than vertical resolution (Zhang et al., 2020, Smith et al., 2018, Jahn and Gallus, 2018)."

3. Line 88: "once" or "one"?

Response: This sentence has been revised for clarity:

"Analyses presented here use model output sampled once hourly (at the top of the hour) for December 2007 to May 2008, and thus consider over 4300 profiles for each grid cell within a sub-domain (D03) comprising 147 by 100 grid cells that encompasses the state of Iowa (Figure 1)."

4. Lines 100-101: ". . .hub height. . . . . .nominal rotor plane. . .". If I understand correctly, there is no wind turbine modeled in the analysis. Presenting wind turbine's terms with no context may confound the reader as to where there is actually a wind turbine involved. I recommend to previously explain this. My personal suggestion would be something like: "Parameters are calculated in a vertical length (from 50 m to 150 m) where a hypothetical wind turbine (not modeled here) may spin, and hereafter we call that span the nominal rotor height, and the height 100 m, the hub height."

Response: A sentence clarifying that no wind turbines are modeled in the study is now included:

"Parameters are considered in the vertical length of 50 to 150 m above ground level (a.g.l.), representing the rotor span of a typical wind turbine (not modelled here) with a rotor radius of 50 m and hub height of 100 m."

5. Lines 110-119: You may prefer to use a consistent style to enumerate a, b, c, d; either all of them in a single paragraph or each one in separated lines.

Response: This has been incorporated into Section 2.2, and now reads:

"The parameters considered are: (a) Mean turbulent kinetic energy (TKE) across the rotor plane derived by the PBL scheme. (b) Wind speed at a nominal hub-height of 100 m a.g.l. (c) The median Richardson number across the nominal rotor plane (RiRotor) specified as 50 – 150 m a.g.l (Eq. 1). (d) Mean shear across the nominal rotor plane (Eq. 2)."

6. Lines 120-121: "All variables ... are computed at a disjunct hourly time step ... Ri_rotor is computed using output disjunct at three hourly intervals." Would you provide more details as to how and why time steps are "disjunct"?

Response: For clarity, the word "disjunct" has been removed. The sentence now reads:

"All variables except RiRotor are computed using output sampled at an hourly time step, while RiRotor is computed using variables output at three hourly intervals."

7. Lines 130-133: "The five values used are 1:1:5 m/s. . . The five thresholds used C4 WESD Interactive comment Printer-friendly version Discussion paper are 10:10:50%." One can infer that you mean "The five values from 1 m/s to 5 m/s in increments of 1" and "The five values from 10 % to 50 % in increments of 10" but the notation may be unclear to many. Is the notation supported by a standard?

Response: The notation has been changed throughout the text and is now described concisely in the following:

"The criteria are grouped into five classes based on strictness and usage in literature, from the least strict (1 ms-1 fixed, 10% variable) to the strictest (5 ms-1 fixed, 50% variable) (Table 2). Threshold strictness increases across groups in increments of 1 ms-1 for fixed and 10% for variable."

8. Lines 168-171: May you rephrase Figure 2 caption? "...during hour identified as exhibiting LLJ..." seems to indicate that the red curve was obtained during a specific, single hour. However, the next sentence ("These profiles are computed for all hourly

profiles from all grid cells...") points to something like an average profile using, not only several cells, but also from several hours. Moreover, I am curious as to how LLJs taking place in different grids and at different hours (and therefore potentially peaking at variable heights) were averaged into a unique profile. One can infer that you selected a specific hour in which calculations showed LLJ happening in several cells, then you combined the normalized profiles from those cells into an average profile (the heights of the LLJ's peaks should be very similar because they are happening in the same hour in not-so-distant locations), and finally did the same with the profiles in the grids with no LLJ happening to obtain the black curve. Is this interpretation correct?

Response: To clarify the method of compositing the wind speed profiles, the figure caption has been edited and a few sentences have been added to improve clarity overall regarding this figure. The sentence describing the calculation now reads:

"The spatiotemporal mean LLJ core wind speed computed using wind speed values across each vertical layer for all hours from all grid cells is approximately 9.55 ms-1 and is centered at about 183 m a.g.l."

And the figure caption for Fig 2 now reads: "Mean wind speed profiles during all hours identified as exhibiting LLJ and those without (non-LLJ). These profiles are computed for all hourly profiles (in the entire time domain from December 2007 to May 2008) from all grid cells and each profile is normalized by the maximum wind speed after compositing. The LLJ detection algorithm uses a variable threshold of 20%. Both mean wind speed profiles are plotted against the temporally and spatially averaged mean height of each vertical level (âŹę)." 9. Line 173: It is important to clarify that this modal value is obtained within the scope of this analysis (which only detected LLJs using wind speeds within the lowest 530 m of the atmosphere, as mentioned in section 2.1) and therefore cannot be interpreted as the modal value representing all LLJs in the region, which should also include LLJs peaking at higher altitudes. The modal value of all LLJs with core at any height would be probably higher.

Response: The sentence has been modified to reiterate the fact that these analyses are dependent on the vertical window considered:

"The modal value of LLJ height in the vertical window considered is between 100-150 m a.g.l. (the upper extent of the nominal rotor plane), and approximately 39% of LLJs have a wind speed maximum within the nominal rotor plane of 50-150 m (Figure 3(b))."

10. Line 177: You may consider to spell out "WS" as "wind speed" if that is what it means. "WS" could also stands for "wind shear", for example.

Response: This has been clarified in the figure description:

"Fig 3. – Probability distributions from a domain-wide sample of all hourly realizations (n=4392) of vertical LLJ wind speed (WS) profiles for: (a) LLJ duration; (b) Height of the jet core. Note that LLJ with durations of over 20 hours were identified, but the distribution is truncated at 20 hours for legibility."

11. Line 192: "see below". You need to be more specific as to where in the text you are directing the reader. Is it to section 3.2?

Response: The reader is now directed to section 3.2: "This is likely due to a complex combination of the following factors; (a) the LLJ selection criteria is more readily met at lower wind speeds (Section 3.2)..."

12. Lines 112-113: "The mean winter flow direction for both LLJ and non-LLJ is westerly," The arrows don't contrast much, but it seems that westerly flow direction is for C5 WESD Interactive comment Printer-friendly version Discussion paper non-LLJs only, while LLJs exhibit much more spatial variability (Figure 5a)

Response: See response in (13).

13. Lines 113-114: "...while easterly flow is more common during the spring months." The arrows don't contrast much, but it seems that that easterly flow direction is for LLJs only, while non-LLJs seem to come mostly from south and southeast (Figure 5b).

Response: Descriptions for flow directions in both winter and spring have been up-dated:

"The mean winter flow direction for both LLJs and non-LLJs exhibits a westerly com-ponent for all grid cells considered, while easterly flow components are more common during the spring months. Rotor plane wind directions during LLJ events exhibit more spatial variability than during non-LLJ events. Springtime LLJs exhibit less spatial vari-ability in wind direction than winter LLJs, coinciding with the increased frequency of winter LLJs compared to spring LLJs."

14. Lines 212-220: Your cross-reference style is not consistent: Line 212: Figure 5(a) and (b). Line 218: Figure 5a. Line 220: Figure 5

Response: Cross-reference style is now consistent.

15. Line 221 (figure 5): Would it be possible to use a more contrasting color for LLJ arrows in subfigures (a) and (b)?

Response: The color for the LLJ arrows has now been updated to a more contrasting color.

16. Lines 222-224: If the color scale represents elevation and wind vectors are repre-sented with arrows, then it is not clear which element in figures 5a and 5b is represent-ing "contours of regions of highest 10% of LLJ frequency".

Response: The figure description has now been updated to indicate what represents the regions of highest 10% of LLJ frequency and now reads:

"Dec-Feb. Regional elevation (m) with contours of regions of highest 10% of LLJ fre-quency (>.26). Average LLJ ( ) and non-LLJ (white) wind vectors at nominal turbine hub height of 100 m; (b) – Mar-May. Regional elevation (m) with contours (black, con-tour values given in white markers) of regions of highest 10% of LLJ frequency (>.19). Average LLJ and non-LLJ wind vectors at nominal turbine hub height of 100 m; (c) – Dec-Feb. Regional mean LLJ duration; (d) – Mar-May. Regional mean LLJ duration.

Black markers indicate wind turbine locations."

17. Lines 277-279: "The median LLJ height is higher by approximately 20 m when the fixed wind speed thresholds are applied than in use of any of the variable thresholds. . ." Revise sentence grammar.

Response: The sentence has been revised and now reads:

"Usage of a fixed threshold extracts LLJs with higher wind speed maxima overall; across all criteria groups, the median LLJ height is higher by approximately 20 m when fixed thresholds are applied (Figure 8(a))."

18. Line 287: "...for applied for..." Check grammar

Response: This sentence has been revised and now reads: "As in results for an individual grid cell shown in Figure 6, as the absolute threshold applied for LLJ detection increases. . ."

Review Responses – Review 2

Major Comments

1. Goal 1: to define a climatology one has to use at least a year of data and preferably more (to capture all relevant mechanisms). The usual definition of a climatological period is 30 years. Also in the context of wind energy the turbine lifetime is generally >20 years. It is quite likely that also summer time jets are quite abundant, if not more, than during winter and spring. For example in the cited paper of Baas et al. (2009), most LLJs were observed during summer. If you don't use a full year of data the paper is just a case study and in that case I don't think it contains enough novelty to publish the results.

Response: Numerous recent WRF-simulated LLJ studies have been for very short time periods, around 24 or 48 hours in total, or investigating a number of 24-hour periods (Squitieri and Gallus, 2016, Tay et al., 2020, Gevorgyan, 2018, Smith et al., 2018).

This work is unique in that the LLJs have been simulated for such a long timeframe (6 months) and with high vertical, horizontal, and temporal resolutions. By comparison, a recent publication investigating the mechanisms behind the formation of the Great Plains LLJ utilized 5 months of WRF simulation data and used 7 vertical layers (Jiménez-Sánchez et al., 2020). Here, two fully simulated seasons were chosen for this analysis to compare regional LLJ/non-LLJ flow directions, and LLJ durations and frequencies for a cool and warm season in the contemporary climate. The seasonal analyses primary novel focus is in defining the LLJ and non-LLJ rotor plane characteristics over long periods rather than focusing on one case study and thereby being able to quantify domain-wide LLJ frequencies and durations. We have also presented sensitivity studies showing how the method of defining LLJ and resolution of wind speed output impact the results.

2. Goal 3: I agree with the paper that the detection could depend on resolution, but I was expecting to see a proposition of a method to help diagnosing the jet independent of resolution. At least something better than linear interpolation should be tested (see comment below).

Response: This has been implemented and shown to work well – the authors are grateful for the interesting suggestion. The following paragraph has been added to explain the process of implementing the polynomial interpolation:

"The usage of a polynomial interpolation to account for lower output resolution when extracting LLJs is shown to reduce sensitivity in LLJ characteristics (Table 6). Winter wind speed output at full resolution is firstly analyzed for LLJs under the 20% variable criterion. From this, wind speed profiles corresponding with identified LLJs are sampled at quarter resolution (resulting in wind speed profiles comprised of 7 vertical layers). A sixth-degree polynomial is then fit to each of these wind speed profiles to extrapolate the non-linear LLJ shape between wind speed values at each layer. After creation of the polynomial, the quarter resolution height AGL for each profile is linearly interpolated to that of the full resolution output (25 layers). These linearly interpolated height values

are then input into the polynomial function for each wind speed profile to extrapolate the quarter-resolution output into full-resolution output. These extrapolated profiles are then input into the LLJ detection algorithm (20% variable) and resulting ensemble characteristics are compared."

And, Table 6 has been added to show results of the interpolation. Sentences in the abstract and conclusion are included to reflect results from the interpolation.

Further, to clarify that no linear interpolation was used between layers in the output of the sensitivity study, the following was added in the text:

"The profiles are not linearly interpolated between vertical layers; the LLJs can only exhibit maxima at heights at the 25, 13, and 7 vertical layers considered (to parallel the extraction of LLJ profiles from observational data in which there are a number of fixed datapoints)."

And in the description of Figure 10: "Note: layers are connected linearly for figure visibility, but the LLJ wind speed maxima can only occur at the heights defined at the vertical layers (25, 13, and 7 heights respectively for each resolution)."

Minor Comments

1. l7: I find it a bit confusing that the abbreviation LLJ is both used to indicate singular and plural. Maybe better to use LLJ for singular and LLJs for plural?

Response: This has been changed throughout the manuscript; all instances of plural low-level jets are now referred to as "LLJs."

2. l27: This is usually referred to as baroclincity, please add that term

Response: The sentence has been rewritten to include baroclinicity, and now reads as:

"Mechanisms commonly invoked to describe the forcing mechanisms include diurnal variations in baroclinicity over sloping terrain (referred to as the Holton mechanism, (Holton, 1967)) and diurnal variations in boundary layer friction (referred to as Black-

adar mechanism (Blackadar, 1957)).”

3. Table 2: It is not really clear to me whether these criteria are used seperately or together. If they are not used together, you should put them in seperate tables.

Response: Table 2 has now been organized into three tables – Table 2 now describes the format of the criteria study and the LLJ frequencies previously included in Table 2 have been moved to Tables 3 and 4 in the Results section. Tables 3 and 4 are now separate tables because the criteria are used separately. We have also written a sentence that clarifies this further in the text:

“Variable and fixed criteria in each group are studied separately to examine the type of LLJ extracted by each unique definition. In other words, in every case, either a fixed or variable criterion is applied; the criteria are not used in tandem throughout the study.”

4. l87: To represent a real climatological study one should at least cover all seasons.

Response: The term “climatology” has been changed to “seasonal analysis” throughout the work.

5. l231: This discussion would be much more interesting with some more physical interpretation. If you plot geostrophic wind speed and thermal wind speed in Fig. 5 it becomes clear if this mechanism plays a role here.

Response: The authors agree this would be an interesting contribution and will look into exploring it in future work.

6. l261: The explanation of this figure confusining and had to read this section several times to understand what was being plotted in Fig. 7. I am I correct that for group 2, approx. 60

Response: Sentences have been added to this section to clarify the results and methods. The figure description has also been modified to improve clarity. The section now reads:

Despite similarity in the frequency with which LLJs are detected as shown in Tables 3 and 4, the two criteria types (even in the least strict criteria group of 1 ms-1 fixed, 10% variable) identify a substantial number of different, distinct LLJ events. For the least stringent criteria group (lowest thresholds), of the total number of times that a LLJ is identified between the two criteria (the intersection of identified LLJ), the criteria extract different LLJ events 20% of the time (i.e. a LLJ is identified by one type of criterion but not the other). Thus, the relative frequency of disagreement is 20%. This relative frequency of disagreement increases to nearly 40% for the second criteria group (2 ms-1 fixed, 20% variable), in which the variable and fixed criteria identify different LLJ profiles 40% of the time (thus they identify the same hourly WS profiles as LLJs 60% of the time) (Figure 7). The frequency with which LLJs are identified by variable criteria but not by fixed, and vice versa, is relatively equal for the first three criteria groups. However, as threshold stringency increases (criteria groups 4 and 5), LLJs are more likely to be identified by fixed criteria than when the variable threshold is applied and the identified LLJ events become more dissimilar, with the two criteria identifying the same LLJ events only 10% of the time (Figure 7). These results indicate that the usage of varying LLJ definitions in literature (a fixed threshold only, or a fixed and variable threshold in tandem) potentially results in frequent identification of entirely different LLJ events."

And the caption for Figure 7 now reads:

"Fig 7. – Relative frequency of disagreement of LLJ identification between analyses using a fixed threshold and a variable threshold. In each criteria group, the variable and fixed thresholds are applied separately to the same hourly wind speed profiles to generate frequencies of LLJ identification for each type of threshold. Bars represent the proportion of LLJ identifications in which one criterion identifies a LLJ while the other does not (the relative disagreement in LLJ identification between fixed and variable criteria). Bars are shaded by the proportion of disagreements in which: a LLJ is identified by fixed criteria but not variable (black), a LLJ is identified by variable criteria

but not fixed (green). Calculated from hourly output from single grid cell with highest LLJ frequency as indicated by the seasonal analysis (see Figure 1 for location)."

7. l294: "differs markedly" -> I can hardly distinguish any differences in Fig. 9. It would be more clear with a difference between the two plots, but also then I would probably not call it a marked difference. It seems it would be 1-2

Response: This sentence has been modified to better reflect the differences between the two figures and now reads:

"For criteria group 2 featuring LLJ definitions commonly used in literature separately or in tandem, (2 ms-1 fixed, 20% variable), the spatial distribution of LLJ frequency is sensitive to the threshold employed, particularly in regions of sloping and complex terrain (Figure 9)."

8. l328-330: This description is not very clear to me, maybe an equation would be better. So you normalize the wind speed profile by a maximum value in each grid cell and then calculate a frequency using the variable threshold and subtract those two frequencies? But then a difference of 0.1 is quite big, so it might be worth putting some more emphasis on that result in panel a?

Response: This has been clarified through the use of additional sentences in the following paragraph, and the result has been further explained to emphasize the larger differences that were found (up to ∼0.16):

"LLJ characteristics (particularly jet core height) are sensitive to the model output resolution but spatial variability appears to be less sensitive. The temporal mean LLJ frequency and duration in each WRF grid cell, as extracted from quarter resolution and full resolution output, are normalized relative to their respective domain-wide maximum values (Figure 11). This process defines the domain-wide variations in LLJ frequency and duration for full resolution and quarter resolution output irrespective of the numerical values of each. The resulting normalized LLJ frequency and durations for both resolutions allow for comparison of spatial variability. Most regions (irrespective of terrain elevation) display low sensitivity to reductions in resolution (Figure 11). Maximum positive and negative differences between normalized frequency and duration range from approximately -0.05 to 0.16, respectively. Regions of maximum spatial variability differences occur sporadically throughout the domain and do not correspond with terrain elevation. Regardless of these areas of high variability difference, the spatial patterns of LLJ frequency and duration are comparatively insensitive to the down-sampling of vertical resolution for most of the domain. Further, regions identified as having the highest frequency and temporal mean duration (the highest 5% of each quantity) of LLJs are similar when the LLJ detection algorithm is applied to output at the original vertical resolution and one-quarter vertical resolution (Figure 11(a)). However, there is more divergence in spatial variation of LLJ duration than frequency when these contours are considered (Figure 11(b)). This potentially indicates that inter-study comparisons of regions of high LLJ frequency (and less so duration) may be possible, even under reduced vertical resolution of observational data and/or model output."

9. Fig. 10. This analysis requires the authors to use a simple polynomial fit or something similar to extrapolate the low-resolution case. Using a linear extrapolation in the points of the wind profile clearly does not reflect the non-linear behaviour of a LLJ profile.

Response: This is addressed in reply to the second major comment.

10. Conclusion: I was expecting to see some discussion on which method would be better or could be more suitable in certain conditions. The paper could benefit from a discussion section at the end of the results.

Response: A deeper explanation of conditions during LLJs extracted with each algorithm has been included as further discussion in the sensitivity study section:

"Higher LLJ speeds in the surveyed region correspond to an atmosphere that is near-neutral and enhanced TKE (Aird et al., 2020). It is possible that a fixed criterion is

more appropriate than a variable criterion to ensure that high speed LLJ are extracted reliably. Shorter duration, higher speed jets with enhanced TKE, such as those observed in higher frequency over complex terrain elevation, are less likely to be captured with the usage of a variable criterion (Figure 9). In contrast, the variable criterion extracts a higher number of LLJ with low-magnitude wind speed maxima and higher duration. The decreased wind speeds of the LLJs captured under a variable criterion likely correspond to more stable conditions and decreased TKE. These characteristic differences further account for the higher frequency of LLJ extracted under a variable criterion in the region of the state with less complex and sloping terrain (Figure 9)."

Please also note the supplement to this comment:
https://wes.copernicus.org/preprints/wes-2020-113/wes-2020-113-AC1-supplement.pdf

---

## Editor Decision (ED1)

2020-113

Abstract
- TKE not defined on first use (always define acronyms on first use)
- LLJ repeated a huge number of times – can this be reduced?
- Can you speak to the importance of LLJ? Why do we care?

Introduction
- Lower-troposphoric doesn't mean much to many people, can you describe things in a bit more of a general way? WES is a domain journal not a discipline journal, so you want to make sure the article is accessible to a broader wind audience that may not be meteorologists
- Same comment goes to why do we care about LLJ for wind? Make sure this is described so that the reader has sufficient context
- Generally the first paragraph has a lot of jargon that is very specific to the meteorological community and not the broader wind community – could be helpful to provide again more plain description, use "in other words, …" to give layman definitions of the key concepts
- On. This same point, it might even be nice to have a few simple graphics illustrating a LLJ or a situation where one is / is not present to contrast
- Lines 49 to 54 speak to the importance of the work – this should be elevated and extended
- After line 72, the paper jumps right to a description of the approach in the paper, but will the paper remedy the limitations around standard characterization that was highlighted in lines 55 to 72? I think it can be strengthened how this paper will address the shortcomings in the earlier work – its in the actual results but doesn't come through super strong
- It may also be helpful to have a paper roadmap as a last paragraph (its sort of there but not really)

Methodology
- Maybe provide a brief description of WRF for the general wind community
- Figure 2 would actually make a nice graphic for the introduction similar to what I mentioned above

Results
- I like figure 7 but I think it would be easier to see actually if it were two separate figures side-by-side so the y-axis could provide an exact description of the content (rather than having it in the caption)
- Figure 8 is a bit hard to digest – there is a lot of information being shared here but the flow is a bit hard to follow in this section moving from the discussion of figure 7 to discussion of figure 8
- You may consider even sub-sectioning section 3.2
- It would be good (since there isn't a separate discussion section) to add a paragraph after figure 9 to summarize the key insights from the analysis in section 3.2

- Can you explain better why you chose the variable threshold of 20% for section 3.3 work? Maybe link it back to the discussion in prior 3.2? It goes back to the usefulness of a transition paragraph between sections 3.2 and 3.3
- Do you think you would expect the same or similar results in section 3.3 if you used different criteria?
- Again, consider a summary / key point / transition paragraph after figure 11 for end of section 3.3

Conclusions
- Conclusions are largely descriptive and repetitive of paper content – can they be more succinct and speak more clearly to the key findings and contribution of the work? what recommendations would you make based on the work in terms of LLJ characterization and identification? What future work remains to be done / where would you go from here?

---

## Author Response (AR2)

**Editorial Review**

The authors greatly appreciate the fantastic and thoughtful feedback in this review and have worked to implement it into the manuscript in the following.

**Abstract**

*TKE not defined on first use (always define acronyms on first use)*

TKE has been defined before it is used:

Nocturnal LLJs are most frequently associated with stable stratification and low turbulent kinetic energy **(TKE)** and hence are more frequent during the winter months.

*LLJ repeated a huge number of times – can this be reduced?*

5 instances of "LLJ" have been removed, and the abstract now reads:

Output from six months of high-resolution simulations with the Weather Research and Forecasting (WRF) model are analyzed to characterize local low-level jets (LLJ) over Iowa for winter and spring in the contemporary climate. Analyses using a detection algorithm wherein the wind speed above and below the jet maximum must be below 80% of the jet wind speed within a vertical window of approximately 20 m – 530 m a.g.l. indicate the presence of a LLJ in at least one of the 14700 4 km by 4 km grid cells over Iowa on 98% of nights. Nocturnal LLJs are most frequently associated with stable stratification and low turbulent kinetic energy (TKE) and hence are more frequent during the winter months. The spatiotemporal mean LLJ maximum (jet core) wind speed is 9.55 ms$^{-1}$ and the mean height is 182 m. Locations of high LLJ frequency and duration across the state are seasonally varying with a mean duration of 3.5 hours. Highest frequency occurs in the topographically complex northwest of the state in winter, and in the flatter northeast of the state in spring. Sensitivity of LLJ characteristics to the: i) LLJ definition and ii) vertical resolution at which the WRF output is sampled are examined. LLJ definitions commonly used in literature are considered in the first sensitivity analysis. These sensitivity analyses indicate that LLJ characteristics are highly variable with definition. Use of different definitions identifies both different frequencies of LLJs and different LLJ events. Further, when the model output is down-sampled to lower vertical resolution, the mean jet core wind speed height decrease, but spatial distributions of regions of high frequency and duration are conserved. Implementation of a polynomial interpolation to extrapolate down-sampled output to full-resolution results in reduced sensitivity of LLJ characteristics to down-sampling.

*Can you speak to the importance of LLJ? Why do we care?*

The following sentence has been added to the abstract:

Low-level jets affect rotor plane aerodynamic loading, turbine structural loading, and turbine performance, and thus accurate characterization and identification is pertinent.

**Introduction**

*Lower-tropospheric doesn't mean much to many people, can you describe things in a bit more of a general way? WES is a domain journal not a discipline journal, so you want to make sure the article is accessible to a broader wind audience that may not be meteorologists*

The following sentences have been improved for clarity and to improve general understanding of the phrases used:

The term low-level jet (LLJ) is applied to any lower-tropospheric (approximately 2 km or below) maximum of horizontal winds that exhibits confined vertical extent (Markowski and Richardson, 2011). LLJs are observed episodically in most regions of the world (Rife et al., 2010; Krishnamurthy et al., 2015). LLJ formation mechanisms and manifestations span a range of scales from synoptic (i.e. mid-latitude cyclones) down to meso- (i.e. weather fronts) and micro-scales (i.e. topographic complexity and day-night surface heating) (Blackadar, 1957; Chen and Kpaeyeh, 1993; Lackmann, 2002; Jiang et al., 2007; Tay, 2021).

*Same comment goes to why do we care about LLJ for wind? Make sure this is described so that the reader has sufficient context*

The following sentence has been added to enhance the description of previous LLJ studies to give more context:

LLJs at and below these altitudes have the potential to impact the wind speed, turbulence, and shear across typical wind turbine rotor planes (Gutierrez et al., 2014; Gutierrez et al., 2017; Nunalee and Basu, 2014; Wagner et al., 2019; Aird et al., 2020; Barthelmie et al., 2020). **Further, LLJs can increase wind farm performance through enhancing wake recovery (depending on atmospheric conditions and jet height), and may reduce wind turbine structural loading if the negative shear region of the jet interacts with the nacelle (Gadde and Stevens, 2021; Guttierez, 2017).** If LLJ speed maxima occur at or near the rotor plane, enhancements in turbulence and shear have implications for **aerodynamic blade** loading and longevity (Kelley et al., 2005).

*Generally the first paragraph has a lot of jargon that is very specific to the meteorological community and not the broader wind community – could be helpful to provide again more plain description, use "in other words, …" to give layman definitions of the key concepts*

In addition to previous edits, the paragraph has been extended to give a more general description of the mechanisms and ideas being discussed:

The term low-level jet (LLJ) is applied to any lower-tropospheric (approximately 2 km or below) maximum of horizontal winds that exhibits confined vertical extent (Markowski and Richardson, 2011). LLJs are observed episodically in most regions of the world (Rife et al., 2010; Krishnamurthy et al., 2015). LLJ formation mechanisms and manifestations span a range of scales from synoptic (i.e. mid-latitude cyclones) down to meso- (i.e. weather fronts) and micro-scales (i.e. topographic complexity and day-night surface heating) (Blackadar, 1957; Chen and Kpaeyeh, 1993; Lackmann, 2002; Jiang et al., 2007; Tay, 2021). Mechanisms commonly invoked to describe the forcing mechanisms include diurnal (day-night) variations in baroclinicity over sloping terrain (referred to as the Holton mechanism, (Holton, 1967)) and diurnal variations in boundary layer friction (referred to as Blackadar mechanism (Blackadar, 1957)). Both mechanisms invoke decoupling of the planetary boundary layer from the surface. In the case of the Blackadar mechanism, this decoupling is due to changes in turbulent mixing associated with day-night stability differences. These stability differences begin at sunset as the boundary layer rapidly stabilizes as the land surface cools, resulting in an inertial oscillation that is conducive to LLJ formation. For the Holton mechanism, the decoupling can be attributed to pressure gradients arising from day-night heating of sloping terrain. Thus, both mechanisms result in a wind speed maximum and indicate LLJs are most frequent under stable conditions and hence at nighttime (Holton, 1967), and in areas with

topographic and/or land cover variability (Parish, 1982). LLJ characteristics, such as frequency, intensity and duration also vary by seasonal and inter-annual timescales (Weaver et al., 2009; Liang et al., 2015).

*On. This same point, it might even be nice to have a few simple graphics illustrating a LLJ or a situation where one is / is not present to contrast*

The following graphic has been added to help describe a LLJ profile as compared to non-LLJ profiles and describe their interaction with the rotor plane, as well as defining the location of maximum wind speed, the jet core:

[Figure]

*Lines 49 to 54 speak to the importance of the work – this should be elevated and extended*

In addition to the previous edits clarifying the impact of the LLJ on wind turbines, the following sentence has been added to emphasize that LLJs are going to be even more pertinent with increasing wind turbine dimensions:

If LLJ speed maxima occur at or near the rotor plane, enhancements in turbulence and shear have implications for aerodynamic blade loading and longevity (Kelley et al., 2005). **As wind turbine heights, rotor diameters, and capacities increase, it is likely that LLJs will interact more profoundly and frequently with the rotor plane, with increasing turbine dimensions resulting in more interaction with the jet core (Barthelmie et al., 2020).**

*After line 72, the paper jumps right to a description of the approach in the paper, but will the paper remedy the limitations around standard characterization that was highlighted in lines 55 to 72? I think it can be strengthened how this paper will address the shortcomings in the earlier work – its in the actual results but doesn't come through super strong*

The following paragraph has been added to discuss insights the paper provides for issues highlighted in 55-72:

Thus, due to frequent variation of LLJ definitions, it is pertinent to examine the types of LLJs (characteristics) that each definition extracts and the agreement between definitions. As LLJs occur due to atmospheric forcing on multiple scales (synoptic, meso, micro), it is possible that their wind speed profiles are a consequence of atmospheric conditions during the time of their generation, and jet profiles

might be more likely to be extracted by certain definitions depending on atmospheric conditions or topography. A greater understanding of jets extracted through definitions used throughout literature can thus reduce uncertainty in future studies and inform choice of definition.

*It may also be helpful to have a paper roadmap as a last paragraph (its sort of there but not really)*

The authors appreciate the suggestion but for now the paragraph remains unchanged in this way, to avoid potentially becoming overly repetitive.

**Methodology**

*Maybe provide a brief description of WRF for the general wind community*

The following sentence has been added to explain the uses of WRF and what it is:

The Weather Research and Forecasting Model (WRF) is a mesoscale numerical weather prediction model that is widely used in wind energy assessment and forecasting applications, such as predicting the impact of climate change on wind power generation and creating wind energy production estimates offshore and onshore (Pryor et al, 2020; Salvação and Soares, 2018, Prósper et al., 2019).

*Figure 2 would actually make a nice graphic for the introduction similar to what I mentioned above*

Figure 2 was used as inspiration for the graphic in the introduction.

**Results**

*I like figure 7 but I think it would be easier to see actually if it were two separate figures side-by-side so the y-axis could provide an exact description of the content (rather than having it in the caption)*

The authors like this suggestion but think it might be easier to compare the proportions of LLJs extracted with each definition given how the figure is currently set up.

*Figure 8 is a bit hard to digest – there is a lot of information being shared here but the flow is a bit hard to follow in this section moving from the discussion of figure 7 to discussion of figure 8*

*You may consider even sub-sectioning section 3.2 - It would be good (since there isn't a separate discussion section) to add a paragraph after figure 9 to summarize the key insights from the analysis in section 3.2 –*

3.2 has now been sub-sectioned and reads more easily, and sentences are added to describe the purpose and reasoning behind each sub-section:

**i) Initial demonstration of sensitivity to LLJ definition**

Any LLJ analysis is naturally dependent on the detection algorithm applied. Thus, a sensitivity analysis is performed using differing LLJ detection thresholds (see Table 2). The impact of selecting different thresholds (five different fixed thresholds ranging from 1 to 5 ms$^{-1}$ in increments of 1 ms$^{-1}$ and five different variable thresholds ranging from 10 to 50% in increments of 10%) is illustrated in Figure 7 for the WRF grid cell that exhibited the highest LLJ frequency in the seasonal analysis (grid cell location indicated in Figure 2). Sensitivity is firstly demonstrated for a single grid cell to concisely prove

sensitivity without confounding factors related to terrain elevation. Domain-wide frequencies are presented in Figure 9 for the most frequently used LLJ definitions and indicate that there is terrain-related sensitivity to the LLJ criteria employed.

**ii) Sensitivity of LLJ definition across entire domain (ensemble sensitivity)**

Ensemble characteristics for LLJs extracted with each definition are analyzed to better understand LLJs extracted with each definition. Domain-wide LLJ frequencies are analyzed for the two most common definitions used in LLJ literature (criteria group 2) and indicate where, in a domain with complex terrain, each type of LLJ (as extracted by the definitions) is likeliest to be extracted.

***Can you explain better why you chose the variable threshold of 20% for section 3.3 work? Maybe link it back to the discussion in prior 3.2? It goes back to the usefulness of a transition paragraph between sections 3.2 and 3.3***

The following lines written for the first round of reviews have been added into the manuscript to clarify:

Results from this sensitivity study inform choice of criterion for the initial study; both criteria types are biased toward certain maximum LLJ speeds and choosing a criterion in the least strict group could result in LLJ wind speed profiles that are hardly differentiable from non-LLJ (as indicated by the lower shear displayed in jets extracted in criteria group 1). Further, criteria group 2 features definitions most relevant to previous studies, and the variable criterion chosen allows for analysis of LLJs that might have been previously undefined through usage of only a fixed criterion (as is common in previous literature).

***Do you think you would expect the same or similar results in section 3.3 if you used different criteria?***

This is an interesting question; it seems that this would depend on the consistency of the LLJ profiles themselves, which would be dependent on shear. The following has been added to the manuscript to answer this question in a thought experiment:

Though a 20% variable criterion is utilized for this sensitivity study, it is possible that usage of a different criterion might affect the results and increase the efficacy of the polynomial fit in resolving lower-resolution LLJ profiles. For example, for higher wind speed LLJs (wind speed maximum > 17 ms$^{-1}$) that are extracted by the fixed criterion, shear across the rotor plane remains relatively constant (Aird et al., 2020). In contrast, LLJs exhibiting lower wind speed maxima as are more commonly extracted by the variable criterion (wind speed maximum between 5 and 11 ms$^{-1}$) exhibit a nearly linear decrease in rotor plane shear with an increase in height A.G.L. These differences are attributed to lower jet core maximum heights for LLJs extracted with variable criteria (Figure 9). Thus, it is possible that extrapolating the LLJ profile from lower-resolution wind speed profiles as extracted from a fixed criterion would prove to be more effective due to more constant shear and higher wind speed maxima.

***Again, consider a summary / key point / transition paragraph after figure 11 for end of section 3.3***

The following has been added to summarize the results of the second sensitivity study and relate the two sensitivity studies, as in the previous comment:

Ensemble LLJ characteristics display sensitivity to the resolution of wind speed profiles, but this can be mitigated through extrapolating the wind speed profile to higher resolution through a polynomial fit. This sensitivity appears to be consistent across the domain and irrespective of terrain complexity, as regions of highest LLJ frequency and duration are preserved when LLJs are extracted from full resolution wind

speed profiles and manually down-sampled wind speed profiles. Though a 20% variable criterion is utilized for this sensitivity study,….

**Conclusions**

***Conclusions are largely descriptive and repetitive of paper content – can they be more succinct and speak more clearly to the key findings and contribution of the work?***

The following sentences have been added to concisely describe and summarize findings:

LLJs as extracted by fixed criteria are predominantly characterized by higher speeds and shorter durations. LLJs extracted by a variable criterion exhibit a higher duration and lower wind speed maxima. In the context of previous work, lower LLJ wind speed maxima as extracted by variable criteria correspond to more stable conditions and decreased TKE, further explaining the increase in LLJ duration. The difference in LLJ types as extracted by each definition correspond to terrain complexity; in the region of the state with less complex and sloping terrain, a higher frequency of LLJs are extracted with the variable criterion. Previous literature implements either a fixed criterion (most common) or a fixed and variable criterion in tandem. Thus, it is possible that for regions with less complex terrain, a variable criterion must be implemented to adequately capture all wind speed profiles with LLJ behavior. The converse is true for employing a fixed criterion: to adequately capture higher speed, shorter duration LLJs such as those that occur more frequently over complex and sloping terrain, it is pertinent to employ a fixed criterion. Thus, the usage of both a variable and fixed criterion to extract LLJs is recommended.

***What recommendations would you make based on the work in terms of LLJ characterization and identification?***

See previous, and the following has been added:

 Based on findings, employing a polynomial interpolation to enrich the number of datapoints in the wind speed profile may prove beneficial in resolving ensemble LLJ characteristics.

***What future work remains to be done / where would you go from here?***

A brief sentence about future work has been added to the second to last paragraph in the conclusions:

Future work to explore the impact of LLJ definitions in offshore conditions is warranted.

**Second Round – Review #1**

The authors greatly appreciate that the reviewer has read and reviewed the revised work, and have worked to implement the comments as follows. The authors are grateful for two rounds of very helpful and thoughtful feedback.

***102-105: The performance of the MYNN scheme with respect to resolution and low-level jets is also discussed in Floors et al. (2013). This could be a good reference to add here as well.***

The authors have added the following sentences clarifying results found in previous studies regarding the MYNN scheme:

Key physics settings in the simulation presented here parallel those used in a similar study of the Orinoco LLJ over South America (Jiménez-Sanchéz et al., 2019); i.e. the Mellor-Yamada-Nakanishi-Niino (**MYNN**) 2.5 (Nakanishi and Niino, 2006) PBL scheme is used, along with the MM5 surface layer scheme (Beljaars, 1995), and the Noah land surface model (Tewari et al., 2004). The MYNN scheme is selected as it has been validated previously for WRF simulations in the Great Plains and **shown to adequately model the PBL height when compared to observations (Zhang et al., 2020)**. Further, **studies of LLJs in the Great Plains** indicate that nocturnal LLJ characteristics may be less sensitive to the scheme employed than vertical resolution; **the MYNN scheme has been shown to have minimal mean absolute error when simulating key jet core conditions, particularly with fine vertical grid spacing and a high model top pressure level such as that utilized in this simulation (50 hPa)** (Smith et al., 2018, Jahn and Gallus, 2018). Note that in all analyses presented herein only wind speeds within the lowest 530 m of the atmosphere are considered. This implicitly limits the detection of LLJs to levels below that height.

*l113: it is fine that the authors do not want to run more simulations and have changed the word 'climatology' to 'seasonal analysis', but I think this aspect should still be made more clear in the abstract and conclusions. When somenone would read the abstract one could easily think all data shown in the paper are general, but in fact they only concern spring and winter.*

The following information has been added to the abstract to clarify that six months of output are considered.

**Abstract:** Output from **six months of** high-resolution simulations with the Weather Research and Forecasting (WRF) model are analyzed to characterize local low-level jets (LLJ) over Iowa **for winter and spring in the contemporary climate.**

*l132: You could also consider using the greek letter alpha for wind shear.*

The Greek letter alpha has been added instead of "shear":

(d) Mean shear ($\alpha$) across the nominal rotor plane (Eq. 2).

$$\alpha = \left( \frac{U_{Z_2} - U_{Z_1}}{Z_2 - Z_1} \right) \quad (2)$$

l300: Shorten caption to something like "As Table 3..."

The following has been added to the figure caption in place of the previous sentence:

**Temporal mean wind speed profiles per group are calculated from LLJ events in Tables 3 and 4.**

Table 5: Please include also half and quarter resolution in the table. The number of levels are already visible in the figure and in the discussion you refer to the 'half', 'quarter' etc.

| | Mean Jet Core Wind Speed (ms$^{-1}$) | Mean Height of Jet Core (m a.g.l.) | Mean LLJ Duration (hours) | % LLJ with Jet Cores within the Rotor Plane | Spatiotemporal LLJ Frequency |
|---|---|---|---|---|---|

| Sensitivity Analysis B: Down-sampling of output | | | | | |
|---|---|---|---|---|---|
| Full Resolution: 25 Vertical Levels | 9.55 | 182.64 | 3.52 | 39.15 | 17.32% |
| 13 Vertical Levels (½ **Resolution**) | 9.18 | 172.89 | 3.35 | 41.83 | 15.12% |
| 7 Vertical Levels (¼ **Resolution**) | 8.53 | 156.43 | 2.98 | 46.95 | 10.75% |

Both resolutions have been specified in the table as shown above.

*l411: I assume you fit the polynomial to the quarter resolution so maybe replace 'these' with 'quarter resolution' or something otherwise it is a bit ambigious what 'these' refers to.*

The following sentences have been revised for clarity:

These profiles (**extrapolated to full resolution from quarter resolution**) are then input into the LLJ detection algorithm (20% variable) and resulting ensemble characteristics are compared **to LLJ characteristics from full resolution profiles and the original down-sampled quarter resolution profiles**.